ecology/evolution

amniotes, convergence, cranial morphology, diet, macroevolution

**Author for correspondence:**
Keegan M. Melstrom
e-mail: keeganmelstrom@gmail.com

# The limits of convergence: the roles of phylogeny and dietary ecology in shaping non-avian amniote crania

Keegan M. Melstrom[1,2,3], Kenneth D. Angielczyk[4], Kathleen A. Ritterbush[2] and Randall B. Irmis[2,3]

[1]Dinosaur Institute, Natural History Museum of Los Angeles County, 900 W Exposition Boulevard, Los Angeles, CA 90007, USA
[2]Department of Geology and Geophysics, University of Utah, 115 S 1460 E, Salt Lake City, UT 84112-0102, USA
[3]Natural History Museum of Utah, University of Utah, 301 Wakara Way, Salt Lake City, UT 84108-1214, USA
[4]Negaunee Integrative Research Center, Field Museum of Natural History, 1400 South Lake Shore Drive, Chicago, IL 60605-2496, USA

KMM, 0000-0002-6313-6146; KAR, 0000-0001-5604-7228

Cranial morphology is remarkably varied in living amniotes and the diversity of shapes is thought to correspond with feeding ecology, a relationship repeatedly demonstrated at smaller phylogenetic scales, but one that remains untested across amniote phylogeny. Using a combination of morphometric methods, we investigate the links between phylogenetic relationships, diet and skull shape in an expansive dataset of extant toothed amniotes: mammals, lepidosaurs and crocodylians. We find that both phylogeny and dietary ecology have statistically significant effects on cranial shape. The three major clades largely partition morphospace with limited overlap. Dietary generalists often occupy clade-specific central regions of morphospace. Some parallel changes in cranial shape occur in clades with distinct evolutionary histories but similar diets. However, members of a given clade often present distinct cranial shape solutions for a given diet, and the vast majority of species retain the unique aspects of their ancestral skull plan, underscoring the limits of morphological convergence due to ecology in amniotes. These data demonstrate that certain cranial shapes may provide functional advantages suited to particular dietary ecologies, but accounting for both phylogenetic history and ecology can provide a more nuanced approach to inferring the ecology and functional morphology of cryptic or extinct amniotes.

# 1. Introduction

The concept of convergence is fundamental to our understanding of biological evolution. Not only does it explain why distantly related organisms sometimes closely resemble each other, the evolutionary 'replicates' resulting from convergence are strong empirical evidence for the importance of natural selection [1]. Morphological similarities are often a result of corresponding pressures associated with an organism's ecological role. Form–function relationships that stem from functional convergence are critical for interpreting the lifestyles and ecologies of species where direct observation is not possible because they are rare, secretive or extinct [2,3]. Cursory comparisons between distantly related taxa show intriguing similarities. For instance, both canids and crocodylians possess elongated snouts and consume a large amount of vertebrate material, which may suggest that similarities in skull shape could be related to diet and its functional requirements, even across groups that are separated by over 300 Myr of evolution [4]. But to what extent do these similarities result in morphospace overlap across a large taxonomic sample? Morphology is subject to a combination of selective, phylogenetic and developmental constraints, so when we claim that organisms have converged, how close is the correspondence and what role do factors such as phylogenetic history play in constraining the degree to which organisms can resemble each other? What are the limits of convergence and can similarities in shape alone be used to generate hypotheses about the dietary ecology of distantly related organisms?

Cranial morphology is extraordinarily variable among amniote tetrapods and its association with extrinsic and intrinsic factors has been explored in groups throughout the clade Amniota [5–15]. Skull shape disparity relates to a myriad of factors, including common ancestry (phylogenetic history), developmental constraint, size, feeding ecology, locomotion and stochastic processes (e.g. genetic drift) [1,16–23]. Studies examining these associations, however, frequently have a narrow phylogenetic focus, with few investigations comparing taxa across major tetrapod clades [2,9,13,24–26]. Not only would more encompassing studies allow the identification of common trends driven by convergence, they also would facilitate the inclusion of extinct organisms that either have no close living relative or exhibit cranial shapes not seen in tested clades. A practical reason for the lack of such studies is the absence of a large number of evenly distributed homologous landmarks across crania from distantly related groups, but recently developed methods are mitigating these issues and allow for scrutiny from a broader range of taxa [14,15,27].

Studies of cranial morphology within amniote subclades (mammals, squamates, turtles, crocodylians and birds) frequently show covariance between ecological factors and skeletal morphology [6,8–11,14,15,28–30]. Analyses of cranial variation in squamate reptiles (i.e. lizards) that employ a range of morphometric techniques repeatedly report convergence in herbivores from distantly related clades [5,6,14]. Similarly, research investigating the cranial morphology of living turtles finds that both diet and habitat contribute to skull shape [29,31]. However, such relationships are not always recovered: targeted analyses of birds, geckos and extinct rhynchocephalians independently found that size and phylogenetic relationships predict skull shape better than feeding behaviour. For example, bird skulls are well-studied examples of convergence in the service of feeding behaviour and diet, but the extent of the signal does not extend to all avian subclades [32]. Clearly, phylogenetic history exerts a primary control in determining morphology for many groups [9,32,33].

The few studies that have compared skull shape across disparate taxonomic groups suggest that dietary ecology and feeding behaviour play a consistent role in determining cranial morphology. Specifically, the mechanical constraints imposed by the particular properties of a diet are reflected in changes in the cranial system [5]. A strong dietary signal is present when crania of avians and squamates are compared across a range of habitats and feeding strategies [14,15]. Correlated shifts in diet and cranial morphology also appear across a range of extant marine tetrapod lineages (mammals: pinnipeds, cetaceans and sirenians; crocodiles: *Crocodylus*; turtles; and the marine lepidosaur *Amblyrhynchus*) [34]. An analysis of functionally analogous structures across living crocodylians and odontocetes highlighted morphological similarities between the long-snouted river dolphin and gharial, two distantly related animals that specialize on similar prey in analogous environments [27]. These observations illustrate how distantly related species may converge on a comparable skull morphology as a consequence of similar dietary and environmental ecologies, setting the stage for analysis of crania across a wide spectrum of clades, dietary ecologies and habitats [5,14,15,29,34]. These repeated patterns of morphological convergence among distantly related animals ultimately may shed light on the ecology of extinct organisms, especially those that lack living relatives or possess skull shapes dissimilar to extant amniotes.

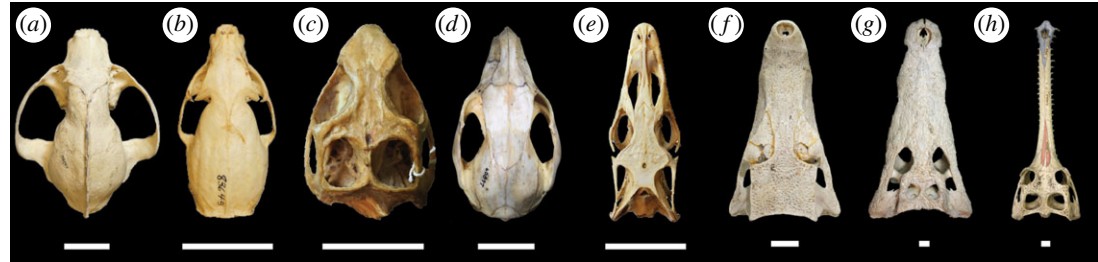

**Figure 1.** Representative non-avian amniote skulls used in this study scaled to the same skull length to emphasize proportional differences. (*a*) *Ailurus fulgens* (FMNH 36755), (*b*) *Galerella sanguinea* (FMNH 83649), (*c*) *Sphenodon punctatus* (UF 11978), (*d*) *Dendrolagus lumholtzi* (FMNH 60897), (*e*) *Varanus salvator* (UF 56879), (*f*) *Paleosuchus palpebrosus* (FMNH 69869), (*h*) *Crocodylus niloticus* (TMM M-1786), (*i*) *Gavialis gangeticus* (NHM R. Unnumbered). FMNH, Field Museum of Natural History, Chicago; NHM, Natural History Museum, London; TMM, Texas Vertebrate Paleontology Collections at The University of Texas, Austin; UF, University of Florida, Gainesville. Scale bar equals 3 cm.

In this study, we investigate broad patterns of cranial shape in living amniotes to test for the limits of morphological convergence. We measure specimens across mammals, lepidosaurs and crocodylians, which encompass a significant amount of the morphological disparity in living tetrapods (figure 1). Our analysis focused on taxa with teeth, given that beaked animals are likely to be subject to different ecological, functional and developmental constraints. We apply a combination of linear and two-dimensional geometric morphometrics (GM) to characterize amniote cranial morphospace and interpret the ecological or evolutionary significance of skull shape within—and between—extant non-avian amniotes. Specifically, we test the hypothesis that there is an overarching relationship between cranial shape and dietary ecology independent of phylogeny, and that distantly related taxa will occupy similar regions of morphospace if they possess the same diet. Furthermore, we seek to identify ecological aspects of cranial shape that are consistent enough to extend to extinct organisms, with the goal of facilitating more nuanced interpretations of dietary ecology and functional morphology.

## 2. Material and methods

### 2.1. Specimen selection

We measured 156 species of non-avian amniotes, including 89 mammals, 20 crocodylians and 47 lepidosaurs. We focused our sampling on adult individuals to avoid issues associated with ontogenetic changes during growth. Taxon selection was based on three primary criteria: reproducibility, variation in morphology and ecology, and clear visibility of landmarks. We sought to sample taxa that were included in previous morphometric studies to facilitate comparison of results, both with geometric morphometric investigations and those quantifying other aspects of shape (e.g. dental morphology) [35–39]. The sampled specimens also display a wide range of cranial morphologies and dietary ecologies. We focused sampling on groups that express broad dietary variation (e.g. carnivorans and iguanids). Among these groups, we preferentially measured and photographed taxa whose dietary information is relatively well documented in the scientific literature and, in some cases, display morphological extremes (e.g. *Corytophanes*) to capture as much variation as possible in a limited dataset.

Some major groups (e.g. birds, turtles, snakes, whales), however, are not represented in this study, primarily due to the absence of homologous points or key landmarks not being visible in either dorsal or lateral view. In particular, primates and cetaceans are excluded from the mammalian dataset. These groups are characterized by extremes in morphology, especially cetaceans due to their extraordinary size and the dramatic modifications associated with a pelagic lifestyle. Previous research has compared the skull shape of crocodylians and toothed whales but used functionally analogous landmarks as opposed to the homologous landmarks that we use here [27]. Similarly, the morphological extremes of primates (e.g. tall crania, forward-facing orbits, etc.) obscure the majority of dorsal landmarks used in this study. In the case of squamates, we did not sample snakes. This group makes up a large proportion of squamate diversity and displays remarkable cranial disparity, including the modification of many sutures (e.g. premaxilla–maxilla) and reduction in maxillary tooth row, precluding the use of a number of our homologous landmarks. Additionally, some snake taxa no

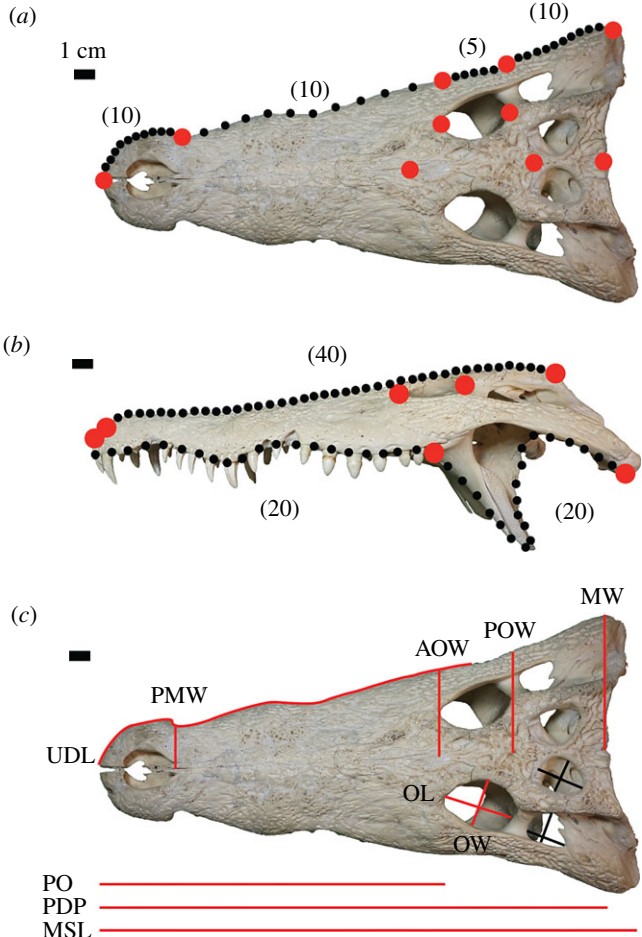

**Figure 2.** The three morphometric measurement systems used in this study illustrated on *Crocodylus niloticus* (TMM M-1786). (*a*) Dorsal geometric morphometric landmarks. (*b*) Lateral geometric morphometric landmarks. (*c*) Linear morphometric measurements. For (*a*) and (*b*), red circles represent landmarks, whereas black circles represent sliding semilandmarks. For (*c*), red lines represent measurements used in all specimens, whereas black lines were only measured in species that possessed these features. Numbers in parentheses refer to the number of semilandmarks used for each curve. AOW, midline width at anterior edge of orbit; MSL, maximum skull length; MW, maximum cranial width; OL, orbit length; OW, orbit width; PDP, premaxilla to the distal edge of parietal at the midline; PMW, midline width at the premaxilla–maxilla suture; PO, premaxilla to the anterior edge of the orbit; POW, midline width at the posterior edge of the orbit; UDL, upper dentition length.

longer possess homologous semilandmark curves in dorsal and lateral views. We did not sample birds and turtles because both groups are edentulous. Although these clades express a broad range of morphologies, habitats and dietary preferences, our decision to focus on toothed amniotes reflects the position of this study in a larger research programme. Specifically, we are interested in developing a multiproxy approach to dietary reconstruction in toothed fossil vertebrates, with a particular focus on fossil crocodylomorphs. Therefore, our taxon sample centres on extant taxa with cranial morphotypes most relevant to this work, while excluding taxa with highly divergent morphologies that are not represented in the fossil record of interest.

## 2.2. Morphological data

We quantified cranial morphologies using a combination of linear and two-dimensional GM (figure 2). We photographed crania using a standardized procedure and limited set of camera lenses to reduce measurement error [40,41]. In sum, we analysed three distinct datasets: a geometric morphometric analysis on the dorsal view of the cranium, a geometric morphometric analysis on the lateral view of the cranium, and a linear morphometric analysis. For the dorsal dataset, we digitized 10 landmarks and 35 sliding semilandmarks, which together capture the broad outline of half of the cranium and key features such as preorbital length, size and location of the orbit, and general morphology of the

postorbital region (electronic supplementary material) (figure 2*a*). Four sliding semilandmark curves separated by landmarks 1–5 collectively outlined the lateral cranial margin. In the left lateral view, seven landmarks highlight key characteristics such as snout length, orbit location, orbit size, length of tooth row and location of important jaw muscle attachments (figure 2*b*). Owing to an absence of repeatable, homologous landmarks on the ventral or dorsal margins in lateral view, 80 sliding semilandmarks separated into three curves quantify the morphology of these regions. Although these methods capture less phenotypic information than three-dimensional GM analyses, well-designed two-dimensional GM schemes detect major morphological patterns [22,42–44]. Importantly, three-dimensional models generated from surface scans or CT data are sensitive to distortion, which can limit any future comparisons to fossil specimens.

We used the multipoint tool of Fiji (V. 2.1) to digitize landmarks and the Bezier Curve tool to digitize semilandmark curves [45]. In Fiji, the multipoint tool generates a set of $X$, $Y$ points for each landmark, which can be easily moved and modified. The Bezier curve tool creates a series of curves that we matched to the precise contours of cranial surfaces. The latter tool can export hundreds of $X$, $Y$ points for each curve and creates high resolution, accurate semilandmark curves for complex surfaces. We then separately saved landmarks and semilandmark curves as .txt files, which were subsequently processed in R and the development environment RStudio (v. 3.6.2; v. 1.2.5033, respectively) [46]. Geometric morphometric data were analysed using the R package 'geomorph' and 'phytools' [47,48]. Sliding semilandmarks were calculated from curves using the function digit.curves, which were calculated based on minimizing bending energy during a Generalized Procrustes Analysis (GPA). Following the GPA, which removes variation in location, orientation and scale, we performed a principal components analysis on the Procrustes coordinates to summarize shape variation and produce a set of uncorrelated shape variables [49–52].

We also investigated shape variation using linear morphometrics and collected 10 measurements using digital callipers (Neiko Tools) and a tailor's measuring tape for curved surfaces (e.g. length of tooth row) or measurements greater than 200 mm (figure 2*c*). We measured specimens directly (i.e. not from photographs) to avoid parallax errors. To ensure overall size did not bias the sample, all measurements were divided by the total cranial length.

## 2.3. Phylogenetic hypotheses

Given the broad taxonomic range of this study, we generated a time-calibrated phylogenetic tree built from a combination of sources (figure 3). The topology for Squamata was based on the molecular dataset of Pyron *et al*. [53], whereas the tree for Crocodylia was based on Erickson *et al*.'s [54] reanalysis of molecular data from Gatsey *et al*. [55]. The mammal topology used a variety of sources for individual groups including carnivorans, marsupials and bats [56–58]. We then built a NEXUS file in Mesquite [59], in which these relatively well agreed-on topologies were combined. Timetree.org, which synthesizes divergence dates derived from molecular data, was used to estimate divergence dates for calculating branch lengths [60]. Subsequent statistical analyses failed in cases of polytomies or unresolved nodes, so we added 1 Myr to a divergence date in these situations. To test the effect of phylogenetic uncertainty, we modified our mammal trees using alternative hypotheses generated primarily from Upham *et al*. [61].

## 2.4. Ecological categories

Amniotes exhibit a remarkable range of dietary ecologies, from extreme specialists (e.g. myrmecophagous mammals and squamates) to broad generalists. We assigned sampled taxa to four coarse dietary categories: carnivore, insectivore, omnivore and herbivore (electronic supplementary material, table S1). Categorizations were obtained from the literature, including large databases and monographic sources (e.g. [6,62–66] and references therein). Although studies with a narrower taxonomic focus often have more precise dietary categories (e.g. terrestrial invertebrates, aquatic invertebrates, terrestrial vertebrates, fish, carrion, fruit, seeds, nectar and other plant material; [14,32,35,67,68]), we used broader dietary groupings to better facilitate comparisons between distantly related groups. This categorization also improves consistency for taxa whose diets and feeding behaviours are more poorly known, which is the case for many squamates.

Following previous studies investigating the relationship between cranial shape and diet, we assigned taxa to a dietary category if they obtain the majority of food from that source [13,14,27,34]. Carnivores are defined as organisms that consume vertebrate material, or large invertebrates (e.g. squid) in the case of

**Figure 3.** Time-calibrated phylogenetic tree of studied non-avian amniotes. Major clades are noted whereas diet is illustrated by terminal branch colour. Grey circles represent 100 Myr. Silhouettes courtesy phylopic.org and Sarah Werning.

marine taxa, for approximately 90% of their diet. Insectivores are categorized as animals that primarily eat (approx. 90%) terrestrial vertebrates. Many taxa have a diverse array of food sources, ranging from insects to plant material. In these cases, we categorized the animal as an omnivore. We designated specimens as herbivores if they consume plant material, such as algae, fruit or grass, for approximately 90% of their food. This categorization method can accommodate herbivores that periodically consume a minor component of animal material (either vertebrate or invertebrate), as well as low- and high-fibre herbivores. Ultimately, diet is flexible and comprises a multivariate set of continuous variables, so the selection of a cut-off of roughly 10% plant or animal matter in specialist categories (i.e. carnivore and herbivore) is arbitrary, but it allows for taxa that periodically consume novel material to remain in their main dietary group.

## 2.5. Statistical analyses

We investigated the covariation of morphological and ecological traits in our three distinct datasets using a combination of statistical tests. To evaluate the amount of shape variation attributable to dietary ecology, allometry and phylogeny in the dorsal and lateral geometric morphometric analyses, we performed a Procrustes ANCOVA using the procD.lm function from the 'geomorph' R package [48]. This function quantifies the amount of morphological variation that can be attributed to one or more factors and is appropriate for use with high-dimensional data (e.g. [14,15,69]). A Procrustes ANCOVA does not take phylogenetic relationships into account and phylogeny was evaluated by categorizing each specimen into one of three groups: Mammalia, Crocodylia and Lepidosauria (electronic

supplementary material, table S1). Similarly, diet was assigned to one of the four categories: carnivore, insectivore, omnivore and herbivore (§3.4; electronic supplementary material, table S1). To account for allometry, we assessed four aspects of size, including Csize (the vector of centroid sizes for each specimen automatically produced by the function gpagen), cranial length, snout length and cranial width. The latter three measurements were derived from the linear morphometric analyses. We then used the procD.pgls function in 'geomorph', which performs a Procrustes ANCOVA in a phylogenetic framework with 9999 iterations to evaluate the effect of dietary ecology and cranial size on the shape of both GM datasets [70]. This analysis tests for the shape variation attributable to diet with time-calibrated phylogenetic relationships explicitly taken into account. Taxa that are closely related will tend to share similar traits. In particular, cranial morphology has previously been shown to contain a significant phylogenetic signal (e.g. [9,32,33,71,72]). We evaluated the degree of phylogenetic signal in shape variables relative to what would be expected under a Brownian motion model of evolution by calculating $K_{mult}$, a multivariate generalization of the $K$ statistic using the function physignal in 'geomorph' [70].

To evaluate the linear morphometric data, we first had to combine continuous linear data with discrete ecological categories (§3.4; electronic supplementary material, table S1). Using the Gower distance metric, a coefficient used to measure the similarity between different samples [49], we separately transformed linear measurements and ecological categories. This step generated a matrix, which permits the comparison of the similarities between mixed categorical (i.e. diet in this study) and continuous data (i.e. morphology). Following this, we performed a principal coordinate analysis (PCO) on each dissimilarity matrix to generate a dataset of continuous variables [67,73]. We used a canonical correlation analysis (CCA) to plot the relationship between morphological and ecological PCO scores. CCA finds linear correlations between two datasets that allow one dataset to maximally predict the second set. In this case, the CCA produces a dataset that allows morphology to maximally predict dietary ecology. Finally, for the linear morphometric dataset, we used the aov function from the 'stats' package to perform an ANOVA to investigate differences in the mean shape across the major phylogenetic groups generated from the CCA data [46].

# 3. Results

In both GM datasets, the first three principal component (PC) axes summarize over 75% of cranial variation, with PC1 accounting for approximately 50% in both cases (figures 4 and 5). PC1 primarily describes variation associated with snout dimensions, with rostrum width and height playing key roles in the dorsal and lateral datasets, respectively. Specimens with exceptionally short snouts plot on one extreme (lowest PC1 scores) and are primarily carnivorous carnivoran mammals, whereas the other morphological extreme (elongate, narrow snouts) is exemplified by *Tomistoma* and *Gavialis*, two carnivorous crocodylians (figure 1). PC2 describes different variation in the dorsal and lateral datasets. In the former, PC2 primarily summarizes the shape associated with the anterior end of the snout and orbital morphology. At negative PC2 scores, the region near the premaxilla is narrow and rounded, whereas the orbits are relatively small. By contrast, orbits are large and the anterior snout is square and relatively wide for positive PC2 scores. In lateral view, PC2 primarily describes variation in the length of the dorsal and ventral cranial margins, with minor input from the relative height of the orbit. The dorsal dataset PC3 describes variation in anterior snout shape, the lateral margin flanking the orbits and the location of the posterior portion of the parietal, whereas the lateral dataset PC3 describes variation in the length of the orbits and the height of the cranium. PC4 and higher axes each account for less than 5% of variance.

Major clades (i.e. mammals, lepidosaurs and crocodylians) largely occupy separate regions of morphospace in the dorsal and lateral GM analyses (figures 4 and 5), with the dorsal dataset best discriminating clades. Only isolated taxa exemplifying morphological extremes (e.g. *Dasypus novemcinctus*, nine-banded armadillo; highest PC 1 score for mammals) plot as outliers of their clade. Mammal and lepidosaur morphospace adjoin in the short-snouted, large orbit region of morphospace, with the carnivorous phocids *Mirounga*, *Erignathus* and *Halichoerus* exemplifying this extreme in mammals (figure 4). In the lateral dataset, crocodylians plot away from the other two major clades, whereas the morphospace of lepidosaurs and mammals overlap to a greater degree than in the dataset of crania in dorsal view, although their ranges remain significantly different (figure 5; Procrustes ANCOVA $p < 0.001$). Aquatic mammals again display extremes in mammalian cranial shape, with select phocids and herbivorous sirenians plotting between a diverse selection of squamates (figure 5).

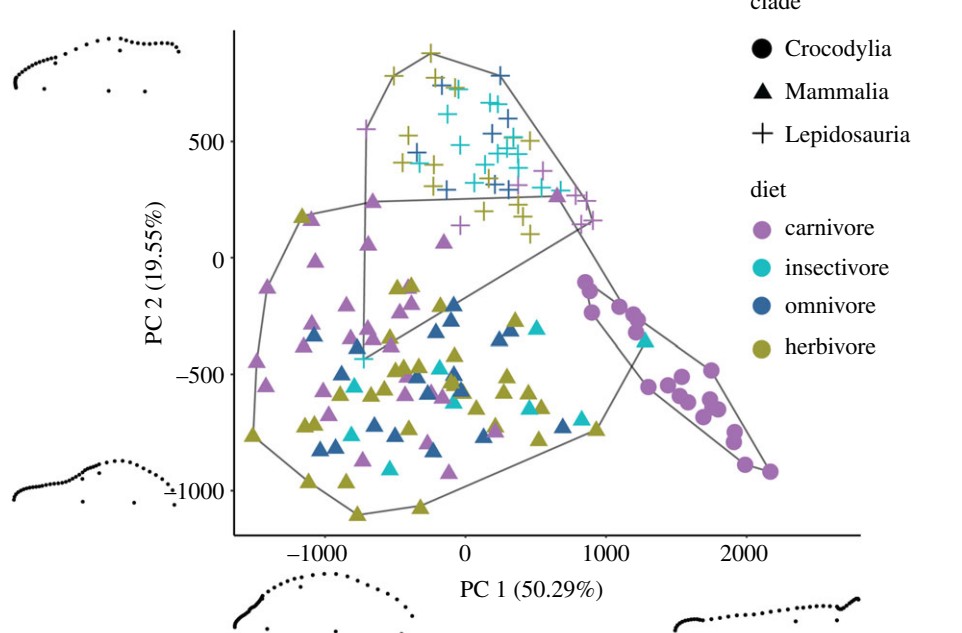

**Figure 4.** Plot of PCs 1 and 2 for the dorsal geometric morphometric dataset. Clades are represented by shapes and diets with colours. The three major clades plot in separate regions of morphospace with relatively little overlap. Convex hulls outline the extent of each clade's morphospace. Extreme morphologies representing the maxima and minima of each PC axis are visualized using landmark plots (anterior to left). Each plot represents the right half of a hypothetical skull.

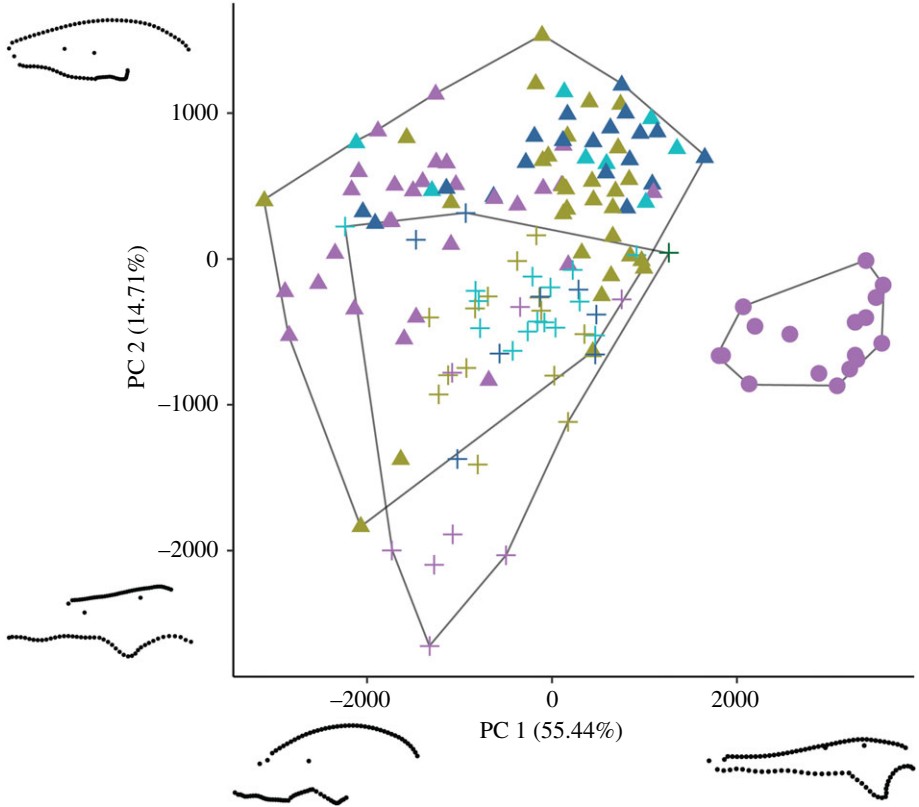

**Figure 5.** Plot of PCs 1 and 2 for the lateral geometric morphometric dataset. Clades are represented by shapes and diets with colours that follow figure 4. Similar to the dorsal dataset, crocodylians plot in separate regions of morphospace, although lepidosaurs and mammals overlap to a greater degree. Convex hulls outline the extent of each clade's morphospace. Extreme morphologies representing the maxima and minima of each PC axis are visualized using landmark plots (anterior to left).

Mammals display a significantly greater amount of morphological disparity (measured as sum of variances) than lepidosaurs or crocodylians in GM analyses (Welch two-sample $t$-test $p < 0.001$), in part driven by the remarkably diverse crania of marine and flying taxa (figures 4 and 5; electronic supplementary material). Despite their low species richness compared to mammals and squamates, crocodylians occupy a relatively large range of morphospace, although it is still the lowest of the tested clades (electronic supplementary material). It is worth noting that these results are an underestimate of total disparity, especially for mammals and lepidosaurs, as we did not include many groups that would certainly exhibit significant shape diversity, such as cetaceans, primates and many groups of limbless lizards.

In the GM datasets, diet and size have a significant impact on cranial morphology when phylogenetic relationships are not explicitly considered (Procrustes ANCOVA $p < 0.05$). In both lepidosaur and mammal datasets, diet is a significant influence on cranial shape, demonstrating this relationship is expressed in both clades with a noteworthy dietary diversity. An allometric relationship is not significant in the dorsal dataset when mammals are evaluated independently ($p = 0.306$), nor in the lateral dataset of crocodylians ($p = 0.116$). This non-significant relationship contrasts with the results of the complete datasets and most other individual clade analyses in both dorsal and lateral view.

When phylogeny is accounted for (i.e. phylogenetic ANOVA), the relationship between diet and cranial shape in the tested GM datasets reveals conflicting assessments (table 1). In dorsal view, this relationship is not statistically significant ($p = 0.071$), whereas in lateral view, it is significant ($p = 0.009$). In both cases, the goodness of fit is low ($R^2 = 0.031$ and $0.042$, respectively), which demonstrates that the predictive power of the relationship is weak. We find that allometry (i.e. log centroid size and log of cranial length) is a significant influence on cranial shape in dorsal view ($p < 0.001$) as well as in lateral view ($p < 0.001$; cranial length only). In a multivariate analysis of covariance, cranial length and diet continue to be a significant influence on morphology in lateral and dorsal datasets ($p < 0.001$). The tests of phylogenetic signal indicate that both dorsal and lateral shapes have less signal than expected under a Brownian motion model of evolution ($K_{mult} = 0.331$ and $0.394$ for dorsal and lateral datasets, respectively). These low $K$ values suggest that morphological variance tends to be distributed more within clades (e.g. carnivorans or iguanids) than between them. In other words, the cranial shapes of closely related taxa tend to diverge more than expected based on their phylogenetic relationships. When alternative phylogenetic hypotheses are tested, phylogenetic ANOVA and strength of phylogenetic signal values remain largely unchanged. These results suggest that our results are driven by broad phylogenetic relationships and not dependent on smaller clade-level patterns.

Within each clade, trends in morphospace occupation with regard to diet are notable. In the GM dorsal dataset, carnivorous squamates plot towards the extremes of lepidosaur morphospace. Carnivorous varanids approach the morphospace of crocodylians, on account of their relatively long snouts, whereas short-snouted carnivores, such as *Heloderma* and *Shinisaurus*, approach the morphospace occupied by carnivorous pinnipeds. Herbivorous lepidosaurs occupy a much larger region, with some plotting near morphospace extremes, but many others overlap with insectivores and omnivores closer to the mean region of morphospace. These patterns are driven by phylogenetic relationships, and when phylogeny is considered trends between diet and cranial shape in both datasets break down ($p = 0.972$; electronic supplementary material).

Among mammals, carnivores tend to have shorter crania with relatively large orbits, although *Canis* and carnivorous marsupials possess relatively long snouts (figures 4 and 5; electronic supplementary material, figure S7). Many herbivorous mammals, including *Equus*, *Macropus*, *Odocoileus* and *Sylvilagus*, independently evolved long rostra. The PC1 values of marsupial herbivores and *Sylvilagus* are similar to those of some herbivorous iguanids (figure 4; electronic supplementary material, figure S7). In the lateral dataset, the trend is even more clear, with mammals displaying a statistically significant relationship between shape and diet even when phylogeny is taken into account (table 1). Terrestrial and aquatic carnivorous mammals spanning a wide range of body sizes are primarily found in the tall, short-snouted portion of morphospace (figure 5). The short-snouted region also hosts some herbivores, like the bamboo specialists *Ailurus* and *Ailuropoda*, as well as some squamates (e.g. the herbivorous iguana *Amblyrhynchus*). Iguanids, herbivorous marsupials and other herbivorous mammals overlap near the centre of morphospace, whereas long-snouted crocodylians fall opposite other carnivores. Together, this results in diverging trends in carnivorous mammal shape space, with herbivorous, insectivorous and omnivorous mammals concentrating near the centre of morphospace.

In the linear morphometric dataset, broad patterns are similar to those of the GM analyses. Long-snouted crocodylians plot in the negative region of the CCA1 axis, with many taxa not overlapping

**Table 1.** Summary statistics of Procrustes ANCOVA and phylogenetic ANOVAs testing the relationship between diet, phylogenetic relationships (Procrustes ANCOVA only) and skull shape using geometric morphometric shape data on dorsal and lateral datasets. Regressions were run using 'geomorph' package. Results of $K_{mult}$ are shown to the right of ANOVA results and represent a separate analysis on the dorsal and lateral datasets that include a time-calibrated phylogeny.

| | DF | SS | MS | $R^2$ | F | Z | p-value | $K_{mult}$ |
|---|---|---|---|---|---|---|---|---|
| dorsal dataset | | | | | | | | |
| Procrustes ANCOVA | | | | | | | | |
| diet | 3 | 0.320 | 0.107 | 0.069 | 3.737 | 3.197 | <0.001 | |
| residuals | 152 | 4.349 | 0.029 | 0.931 | | | | |
| total | 155 | 4.669 | | | | | | |
| phylogeny | 2 | 2.131 | 1.065 | 0.456 | 64.183 | 8.124 | <0.001 | |
| residuals | 153 | 2.539 | 0.017 | 0.544 | | | | |
| total | 155 | 4.690 | | | | | | |
| log skull length | 1 | 0.451 | 0.451 | 0.096 | 16.443 | 4.301 | <0.001 | |
| residuals | 154 | 4.219 | 0.027 | 0.905 | | | | |
| total | 155 | 4.67 | | | | | | |
| phylogenetic ANOVA | | | | | | | | |
| diet | 3 | 0.002 | 0.0006 | 0.031 | 1.639 | 1.503 | 0.071 | 0.331 |
| residuals | 152 | 0.058 | 0.0004 | 0.969 | | | | |
| total | 155 | 0.060 | | | | | | |
| log skull length | 145 | 0.058 | <0.001 | 0.972 | 2.3805 | 2.8118 | 0.0004 | |
| residuals | 10 | 0.002 | <0.001 | 0.028 | | | | |
| total | 155 | 0.060 | | | | | | |

(Continued.)

**Table 1.** (Continued.)

| | DF | SS | MS | $R^2$ | F | Z | p-value | $K_{mult}$ |
|---|---|---|---|---|---|---|---|---|
| lateral dataset | | | | | | | | |
| Procrustes ANCOVA | | | | | | | | |
| diet | 3 | 0.300 | 0.100 | 0.042 | 2.129 | 1.806 | 0.037 | |
| residuals | 145 | 6.813 | 0.047 | 0.958 | | | | |
| total | 148 | 7.113 | | | | | | |
| log skull length | 1 | 0.323 | 0.323 | 0.045 | 6.987 | 2.991 | <0.001 | |
| residuals | 147 | 6.791 | 0.0462 | 0.95463 | | | | |
| total | 148 | 7.114 | | | | | | |
| phylogeny | 2 | 2.788 | 1.394 | 0.392 | 47.038 | 7.014 | <0.001 | |
| residuals | 146 | 4.326 | 0.030 | 0.608 | | | | |
| total | 148 | 7.114 | | | | | | |
| phylogenetic ANOVA | | | | | | | | |
| diet | 3 | 0.003 | 0.001 | 0.042 | 2.112 | 2.352 | <0.001 | 0.394 |
| residuals | 145 | 0.075 | 0.001 | 0.956 | | | | |
| total | 148 | 0.078 | | | | | | |
| log skull length | 1 | 0.003 | 0.003 | 0.035 | 5.378 | 3.356 | <0.001 | |
| residuals | 147 | 0.075 | <0.001 | 0.965 | | | | |
| total | 148 | 0.078 | | | | | | |

with either mammals or lepidosaurs. Most lepidosaurs have positive values on the CCA2 axis, separated from mammals and crocodylians. This includes varanids, with relatively long preorbital regions, teiids and some agamids, including the herbivorous *Uromastyx*. In spite of these patterns, morphological overlap is greatest in the linear dataset, although taxonomy remains a significant influence (ANOVA $p < 0.001$; electronic supplementary material, figure S5). The overlapping morphospace is, in part, driven by short-snouted crocodylians (*Paleosuchus* and *Osteolaemus*) approaching the region occupied by long-snouted mammals (*Equus* and *Odocoileus*). More prominently, however, carnivorous marine mammals plot within the same morphospace region as many iguanians. These patterns result in dietary ecology having a significant impact on skull morphology (ANOVA, $p < 0.05$), but this is likely driven by the relatively long snouts of many carnivorous mammals, varanids and crocodylians.

# 4. Discussion

## 4.1. Morphospace occupation and the limits of convergence

This work reveals the complex interplay of phylogenetic history, dietary preferences, allometry and functional demands in determining amniote cranial shape, and illustrates how this interplay limits morphological convergence in non-avian amniotes. Major clades primarily plot in discrete regions of shape space and the data clearly demonstrate that modern crocodylians, mammals and lepidosaurs have partitioned morphospace with only minor overlap. This partitioning is especially evident for crocodylians, which are distinct from nearly all other taxa in both GM datasets, but it is also visible for mammals and lepidosaurs, which show limited overlap (figures 4 and 5; electronic supplementary material). Even in the linear dataset, which captures the least amount of detail, morphological overlap is limited to central regions of shape space. Although high-level phylogenetic patterns provide a basic structure to morphospace occupation, smaller clades within mammals and lepidosaurs exhibit large amounts of morphological variation demonstrating that cranial shape can be dramatically modified within the bounds of the basic skull architecture of the larger clade. This result is consistent with the tests of phylogenetic signal, which show a high level of within-clade shape variation. In all analyses, much of the variance captured relates to rostral morphology, which ranges from short, wide snouted mammals to the elongate, thin snouts of crocodylians (figures 1 and 4). Functional constraints, in particular, appear to drive a noteworthy degree of variation in the mammal dataset. Taxa that spend significant proportions of their life in marine environments (e.g. pinnipeds, sirenians) or those that fly (i.e. chiropterans) occupy the extremes of morphospace. This suggests that the occupation of these environments requires significant cranial modification. Similarly, the ecological role of ambush predator inhabiting the land–water interface plays a role in crocodylian cranial morphology, with their relatively narrow, flattened skulls that are adapted for high bite forces in aquatic conditions [54]. This result is consistent with that of previous work, in which snout shape plays a predominant role in cranial variation, regardless of the clade investigated [6,8,11,12,26,74].

Despite the morphological range covered by the sampled taxa, there are noteworthy regions of morphospace that are unoccupied. This is best exemplified by the regions around crocodylians. In both GM datasets, crocodylians are separated from mammals and lepidosaurs by a considerable distance, with the single exception of *Dasypus* in the dorsal dataset (figure 4; electronic supplementary material, figure S7). In this case, these differences are driven by the exceptionally long snouts and well-developed pterygoids in crocodylians. These unoccupied regions may be, in part, due to sampling, although there are reasons to be sceptical of this hypothesis. For example, the morphological differences driving the separations we observed are not present in unsampled taxa. Although marine mammal cranial shapes tend to fall in extreme areas of mammalian morphospace and are sometimes compared to those of crocodylians because of their long snouts, no marine mammal has enlarged pterygoids comparable to those of the crocodylians in the dataset, regardless of snout length. Similarly, it is unlikely that snakes, if included, would occupy this region. Their relatively short snouts, large orbits and short tooth row would likely result in a distinct placement away from crocodylians. Previous work that found convergence between phylogenetically distant groups used functional landmarks or linear measurements, including those of dental elements [27,34]. These data, as well as the linear analyses here, contain less morphological information, which may facilitate the overlap found in those studies. For these reasons, we predict that these unoccupied regions reflect reality rather than unsampled taxa. Nevertheless, the inclusion of fossil taxa may fill in parts of the currently empty regions of morphospace. Extinct crocodylomorphs exhibit a wide range

of cranial morphologies [75,76] that may span the morphological range from living crocodylians to mammals, as well as the many groups of marine saurians, which frequently exhibit narrow, long-snouted morphologies. Understanding the constraints preventing greater overlap in extant groups may be a fruitful pursuit for future studies because they may offer insights into the morphological limits of each clade.

## 4.2. The effects of diet

Dietary ecology has a significant influence on cranial shape in the sampled non-avian amniotes in the lateral dataset but not in the dorsal dataset in the phylogenetic ANOVA analyses. Additionally, we do not find an overarching significant predictive relationship between these variables that universally applies across all three clades. Instead, members of the three major clades evolve a variety of cranial shapes associated with a given dietary ecology that reflects their idiosyncratic combinations of skull architecture, dental morphology and degree of emphasis on oral processing of food, corroborating findings for a large sample of avians [15]. This result underscores the importance of phylogenetic scale in form–function investigations. Investigations of individual clades often demonstrate more distinct relationships, with dietary ecology playing a key role in shape [11,13,24,25]. By contrast, when a wider phylogenetic focus is applied, such as in this study, phylogeny tends to overprint trophic patterns, although ecology may continue to play a significant, but less predictive role, like that observed in many other analyses and here [6,9,14,15,32,33]. In spite of this result, some degree of convergence is possible, but taking a broad phylogenetic perspective elucidates how general functional principles and specific dietary and other demands, such as allometry, map onto particular cranial architectures to produce similar or different skull shapes [77].

For example, some lepidosaur and mammal carnivores independently evolve relatively short snouts. This morphology reduces the outlever of the mandible, providing a higher mechanical advantage and higher bite forces, especially when coupled with coordinated changes in muscle architecture, as well as a snout shape better able to resist the resulting stresses [11,26]. However, lepidosaurs do not appear to be able to achieve the level of short-snoutedness observed in some mammals, likely reflecting underlying differences in the basic construction of mammalian and lepidosaur crania, such as (but not limited to) the strong constraints incurred by cranial kinesis. Not all carnivores optimize bite force, though. Sampled varanids buck the trend of short-snouted carnivores and approach crocodylians in terms of relative snout length. This group is noteworthy for having exceptionally weak bite forces for carnivores, suggesting different predation tactics [78,79]. This stands in marked contrast with both the shorter-snouted carnivores and long-snouted crocodylians, which optimize their skulls for high bite forces [7,27,54], albeit in different ways. Thus, varanids represent both a unique morphology and approach to predation in our dataset, representing how there are many morphological solutions associated with similar dietary ecologies (i.e. many-to-one mapping [77]).

Among herbivores, an elongate skull evolves independently at least four times in mammals in this dataset alone: *Equus*, *Macropus*, *Odocoileus* and *Sylvilagus*. *Equus* and *Macropus* are grazers that consume large amounts of grass in their diet and their long snouts likely reflect the tendency for mammalian high-fibre herbivores to have batteries of complex teeth suited for extensive oral processing of food (figure 4; [80–83]). The dorsal dataset PC1 values of some marsupial herbivores and *Sylvilagus* also overlap with those of many herbivorous iguanids (i.e. *Ctenosaura*, *Cyclura*, *Iguana*) indicating broad similarities in snout morphology (figure 4). This convergence may represent a reptilian version of an emphasis on increased oral processing, given that iguanids possess a greater number of teeth relative to most other lepidosaurs [84]. It should be noted, however, that some omnivorous and insectivorous mammals and squamates also fall into this range, and there is less overlap in the lateral view dataset.

Nevertheless, there is variation in the morphofunctional solutions herbivores adopt in association with their specialized diet. Herbivorous carnivorans (*Ailurus* and *Ailuropoda*) and some marsupials (*Vombatus*) possess relatively short snouts, likely for consuming tough food. Optimizing the skull for producing strong bite forces to consume vegetation that is similar in hardness results in similar skull morphologies in these distantly related mammalian taxa [26,85,86] and represents a different adaptive syndrome for herbivory than seen in the longer snouted taxa noted above. Some squamate herbivores also have relatively short snouts, which also have been hypothesized to stem from selection for stronger bite forces [6]. These taxa do not reach the extremes of short-snoutedness observed in mammals, such as *Ailuropoda*, again highlighting likely differences in architectural constraints in mammalian and lepidosaur skulls. In particular, fundamental features such as kinetic skulls in

squamates, secondary palates in mammals and crocodylians (although these evolved independently), and diphyodonty in mammals likely exert limitations on morphological variation. Omnivores and insectivores, regardless of clade, consistently overlap in morphospace. Convergence between these ecological groups is also seen in dental complexity, despite differences in gross morphology [36,37]. This overlap could be due to similar pressures placed on taxa that have relatively flexible dietary habits because their foods are typified by a wide range of physical characteristics. This pattern is exemplified by closely related canids, with *Canis aureus*, an omnivore, occupying a cranial shape closer to central regions of morphospace than its close carnivore relative, *Canis lupus* (figure 4; electronic supplementary material, figure S7) [87]. Additionally, omnivorous and insectivorous taxa plot near the centre of their respective clade's morphospace, suggesting less extreme morphologies.

Because of these complexities, coarse dietary categories are not a significant predictor of cranial shape in our dataset, similar to other studies that investigate phylogenetically broad datasets [14,15]. For instance, the herbivore category encompasses specialists on marine algae, grasses, fruits and eucalyptus leaves as well as those that have a much greater flexibility in their herbivorous habits. Different specialists (such as the bamboo-feeders of [26]) will experience divergent selective pressures when consuming their food, which will also vary from the pressures experienced by generalists, and different functional solutions have been recruited in each case. Mapping those varying selective pressures and functional solutions onto different skull architectures further complicates the relationship between herbivory and cranial shape. The coarseness of these categories, however, may be a necessity to compare disparate groups [6,14,64], although high-resolution studies on avian crania generated similar results, despite finely partitioning dietary ecology into a greater number of categories [15,32].

Previous work on marine tetrapods concluded that trophic convergence drove morphological convergence, with distantly related taxa with comparable dietary ecologies displaying similar skull morphologies [27,34]. This overarching relationship was not recovered when tested with a broad sample of non-avian amniotes. Sampled marine mammals often pushed the extremes of mammalian morphospace and approached squamates, but the broad phylogenetic division of morphospace remained the primary control on large-scale skull shape patterns, illustrating a limit to morphological overlap and convergence.

## 4.3. The effects of sampling

The sum of variance measurements demonstrate that mammals exhibit the greatest amount of morphological variation among the sampled groups, followed by lepidosaurs and crocodylians, respectively. Although only a subsample of total diversity, we predict that the ranking of clades found here reflects reality and the inclusion of additional specimens likely will not significantly change disparity trends. We sampled many of the ecological extremes exhibited by these groups, and among mammals the unsampled clades (e.g. Cetacea) include highly divergent cranial morphologies that would only further increase mammalian disparity. Arguably our exclusion of snakes removes a set of extreme squamate morphotypes from our dataset, but we do not think that their inclusion would be sufficient to overturn the overall pattern. Moreover, in our dataset, mammals exhibit the greatest number of occupied ecological roles, varying from fully marine taxa to those with powered flight, as well as the widest size range. Lepidosaurs, despite a higher species richness are primarily limited to small-bodied predator roles, although there are noteworthy exceptions (e.g. varanids, iguanids) [64]. Similarly, crocodylians today are largely limited to a semiaquatic ambush predator role. The narrow ecological range of saurian groups explains, in part, their lower morphological disparity. When squamate taxa deviate from the common small-bodied insectivore category, they tend to explore novel regions of morphospace. Varanids, in particular, exemplify this pattern and are the sole large-bodied terrestrial predator in our squamate dataset, displaying morphological extremes in all morphometric analyses.

Form–function relationships in living taxa are frequently used to predict the ecologies of fossil taxa (e.g. [24,25,29,67,88–90]), based on the assumption that convergent morphologies indicate convergence in diet or other aspects of ecology. The weak association between cranial shape and dietary ecology across clades in our dataset indicates that such comparisons must be carefully considered. Here, we demonstrate that phylogenetic scale is a critical factor when testing the effects of ecology on morphology, with broadly sampled datasets potentially obscuring patterns observed in smaller constituent clades. Additionally, attention must be paid to likely phylogenetic constraints and the fact that multiple solutions exist to a given ecological problem (e.g. herbivory) [77]. Nevertheless, these

broader studies are valuable in identifying the extent of interplay between distinct morphologies that are the legacy of phylogenetic history and the convergent morphologies resulting from common selective pressures. Plotting extinct taxa in a phylogenetically broad morphospace, such as ours, also has the potential to offer important insights, particularly when used in concert with additional morphological (cranial and postcranial), palaeoenvironmental or extrinsic characters (e.g. stomach contents). For example, our dataset shows that high-fibre herbivores that emphasize chewing and durophagous herbivores that specialize in hard food items occupy different areas of morphospace. Depending on where an extinct taxon falls, the results can elucidate both dietary ecology and the functional ways in which the animal is approaching that diet.

## 5. Conclusion

Our results highlight the complex partitioning of cranial morphospace that has occurred in extant amniotes. Sampled crocodylians, lepidosaurs and mammals occupy largely separate areas, but mammals and—to a lesser degree—squamates have extensively radiated within their respective regions. Dietary ecology remains a significant influence and some parallel morphological trends are observed in mammals and lepidosaurs. Carnivorous taxa and hard-object-feeding herbivores independently evolve a short-snouted cranial shape, whereas high-fibre herbivores frequently develop much longer skulls, but in both cases, these patterns are superimposed on a specific clade's underlying skull architecture. The apparent limits on skull shape convergence in amniotes are a double-edged sword when making inferences about extinct or poorly known extant species. Broad generalizations are difficult, but detailed comparisons with neighbouring taxa have the potential to provide insights about potential function and ecology (e.g. differentiating durophagous from high-fibre herbivores), and the trends observed here can be used to infer the dietary ecology of extinct taxa when applied judiciously.

Data accessibility. Linear morphometric data are available in electronic supplementary material [91] and photographs are available in the Dryad Digital Repository: https://doi.org/10.5061/dryad.0k6djh9xc [92].

Authors' contributions. K.M.M. and K.D.A. collected the data. K.M.M., R.B.I and K.A.R. conceived the study and designed the analyses. All authors prepared the manuscript.

Competing interests. The authors have no competing interests.

Funding. This research was funded by a US National Science Foundation Graduate Research Fellowship, and grants from the Geological Society of America, University of Utah, Natural History Museum Los Angeles County, American Museum of Natural History, and Field Museum of Natural History (to K.M.M.).

Acknowledgements. We thank Nefti Camacho (NHM), Adam Ferguson (FMNH), Bruce Patterson (FMNH), Alan Resetar (FMNH), Chris Sagebiel (TMM), Coleman Sheehy (UF), Lauren Smith (FMNH), Carol Spencer (MVZ) and Shannen Robson (NHMU and NHM) for access to specimens. We are also grateful to Ryan Felice, Jacqueline Lungmus, Jonathan Mitchell and Zackary Wistort for advice with R. We greatly appreciate the comments provided by the editor and four reviewers that improved the manuscript significantly.

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
