## [Peer Review File · Royal Society Open Science]

Review History

RSOS-202145.R0 (Original submission)

Review form: Reviewer 1

Is the manuscript scientifically sound in its present form?

No

Are the interpretations and conclusions justified by the results?

No

Is the language acceptable?

Yes

Do you have any ethical concerns with this paper?

No

Have you any concerns about statistical analyses in this paper?

Yes

Recommendation?

Major revision is needed (please make suggestions in comments)

Comments to the Author(s)

Reviewer comments

The manuscript is an interesting and creative effort on understanding major patterns in amniote skull morphology. I applaud the creativity and vision of the authors to address the issue and to generate comparable quantitative data from disparate animals. The introductory section is remarkably clear and well written, and so is the discussion.

However, in its present form the manuscript is not yet suitable for publication.

My main concerns have to do with sampling and methodology. Regarding sample, I think the authors can do a better effort to justify the choosing of taxa. I find the exclusion of beaked amniotes poorly elaborated. Clearly, having “heavily modified skulls” could apply also for many toothed amniotes, such as vipers, cetaceans, tapirs, dinosaurs, rodents, dugongs, varanids, etc. In addition, the sample is markedly counterfactual when considering amniote evolution, given it is composed of some toothed extant amniotes and excludes not only any fossil taxa but also two major amniote lineages such as birds and turtles. The use of the term “non-avian amniote” is indeed inadequate. In any case, I wonder if an extra emphasis on developing landmarks other than based on teeth could have circumvented this issue and allowed to include some birds and turtles (for example, using the landmark 5 in lateral view to depict also the caudalmost border of the beak, but this is only a suggestion). I do not mean by this to ask the authors to expand the sample and repeat the calculations performed, which are really interesting and have a remarkable heuristic value. Any possible sample will have intrinsic advantages and disadvantages. I only want to stress out that authors should be way more cautious when discussing broad evolutionary implications of their results, given they are representing a quite reduced disparity of cranial morphologies. In several places the authors made assumptions or predictions on how the patterns they found will stand if more lineages were included in the sample, which I find is a bit speculative because authors risk hypothesizing beyond the scope of their results.

Concerning calculations, I consider that lacking of allometry tests is a major flaw of the analysis. This can be viewed even as a missed opportunity of further examination of the data, given that the authors have had all the elements needed to perform allometric regressions, which can be done straightforwardly from the data using the same software and almost without extra computing effort. Justification of not testing influence of size on the shape and ecology is poor and entirely inadequate, in my view. In addition, without testing allometry, all results displayed should be considered “preliminary”, so testing for allometry could be considered mandatory in this context. It can be the case, for example, that snout elongation has an allometric component, and this could be being completely overlooked. Insectivory is also constrained by body size. I strongly encourage authors to perform allometric regressions on the sample data, to include them as supplementary information, and to discuss it along the manuscript.

Minor improvements could imply explaining more about the calculations, how they were made, which options were chosen (if Euclidean distances, or Bending Energy, was used for sliding semilandmarks, how models -fits- were constructed for ANOVA or what type of SS was computed, for example). The issue is especially important for the ANOVA and how the models were set, $\text{shape} \sim \text{diet}$, $\text{shape} * \text{diet}$, etc. R commands accept many different configurations for each analysis, which have impact on results obtained, therefore clarifying this can be very useful for understanding adequately the methodology. Table 1 appears to be mixing Blomberg's K and phylogenetic ANOVA's results, which render it confusing.

Some of these comments are also explained in the attached pdf file (see Appendix A), along with some minor suggestions.

Best regards

Review form: Reviewer 2

Is the manuscript scientifically sound in its present form?

Yes

Are the interpretations and conclusions justified by the results?

Yes

Is the language acceptable?

Yes

Do you have any ethical concerns with this paper?

No

Have you any concerns about statistical analyses in this paper?

No

Recommendation?

Accept as is

Comments to the Author(s)

This is a well-designed study that is very well-presented and written, uses appropriate methodological approaches, and which interprets its results carefully and appropriately in the context of previous work. The authors use geometric morphometrics to explore evolutionary convergence in the skulls of modern mammals, crocodylians and lizards, concluding that phylogeny is critically in defining morphology at the scale of major clades, but that patterns within clades are more complex, and result from an interplay of phylogeny and ecology. I can identify no significant issues with this manuscript and think it could be published without revisions. My only real question was around whether some of the gaps in morphospace between major clades might disappear if fossil taxa were included, and so are perhaps the result of extinction. I am particularly thinking of the much more diverse skull morphologies present in fossil crocodylomorphs, which include short-snouted forms. While I do not think the authors necessarily need to modify their paper, a brief consideration of this point in the discussion might be worthwhile.

Review form: Reviewer 3

Is the manuscript scientifically sound in its present form?

Yes

Are the interpretations and conclusions justified by the results?

Yes

Is the language acceptable?

Yes

Do you have any ethical concerns with this paper?

No

Have you any concerns about statistical analyses in this paper?

Yes

Recommendation?

Major revision is needed (please make suggestions in comments)

Comments to the Author(s)

This is a neat study that provides some interesting and thought-provoking results that will increase our knowledge of phylogenetic constraints and drivers of amniote cranial morphology. I can imagine that this manuscript will be highly cited by morphologists and palaeobiologists, including by myself.

I have uploaded my comments onto the main text PDF (see Appendix B) using the sticky note tool as I find this much better than writing out all my comments. In most cases my comments and questions are pretty minor and can be easily addressed or answered by the authors.

The only areas of 'concern' that I want to mention regard the time-scaling of the supertree and the phylogenetic analyses employed. I should stress that I do not believe what the authors have done is incorrect, I just have a few questions for more information/clarification and to make a few suggestions about how these analyses could be added to in order for the discussions and interpretations to be made (even) more robust.

1) I'm more familiar with stratigraphic ways of time-calibrating supertrees, so if the below is not 100% relevant I apologise. You mention throughout the importance of identifying phylogenetic signals in your dataset to interpret phylogenetic constraints in skull morphology and convergence. How come you then use only one time-scaling method? Phylogenetic signals can be influenced by branch lengths and inferred divergence dates and therefore it may be prudent to use an additional independent (or two if possible) time-scaling method and test for phylogenetic signal. Examples employed by many palaeobiologists include FDB, hedman, mbl and cal3. With the exception of FDB these time-scaling methods can be easily done in R. If there are similar levels of phylogenetic signal present in a second time-scaled tree then this would show your results are not a false positive from your chosen time-scaling method.

2) You mention in your results: "When alternative phylogenetic hypotheses are tested the results are not significantly impacted, suggesting the broader phylogenetic relationships are driving this pattern" but don't elaborate further. Where are the results for these alternative phylogenies? I understand that you used some different mammal relationships from Upham et al. but did you use completely different supertrees when testing for alternative phylogenies? You're not incorrect in the way that have resolved polytomies by adding 1 million years to such polytomies and to unresolved divergence dates, but say you had a three way polytomy with taxa A, B and C, how did you decide which taxon of the three diverges first? The best way to account for polytomies and unresolved nodes would be to create 10-20 randomly resolved versions in R (using one of the time-scaling methods mentioned in point 1) and perform the phylosignal analysis (just the Kmult or lambda statistic, see point 3) on all of them. If the trees have the same strength of phylogenetic signal then you can proceed with one tree (i.e. the one of you already have) for all other analyses.

If you decide that this way of resolving phylogenetic uncertainty is not 100% relevant/applicable to your manuscript, that is fine, but at the very least please provide more clarification about how many trees you used and the results from them (supp. Info would be fine).

3) The lambda statistic is also good for deducing levels of phylogenetic signal in a dataset (Munkemuller et al. 2012; Meth. Ecol. Evol. 3, 743-756). This can be performed using the phylosig package and function of the same name in R. This would provide an independent test of whether

PC 1 and PC 2 values for the dorsal and lateral data exhibit phylogenetic signals for the entire dataset.

I want to stress again that I do not think the way you constructed your time-scaled tree is incorrect, I just think because the manuscript hypotheses and analyses hinge on phylogeny, I just want to make sure the methods and outputs are as robust as possible. The only reason I have recommended major revisions is so that you have enough time to explore the additional/alternative methods I recommend. I really want to see this manuscript published and hope that my suggestions can help.

Review form: Reviewer 4

Is the manuscript scientifically sound in its present form?

Yes

Are the interpretations and conclusions justified by the results?

Yes

Is the language acceptable?

Yes

Do you have any ethical concerns with this paper?

No

Have you any concerns about statistical analyses in this paper?

No

Recommendation?

Accept with minor revision (please list in comments)

Comments to the Author(s)

Dear Authors and Editor,

This manuscript investigates the links between shape, diet and phylogeny in extant amniote skulls. The link between diet/shape has often been used somewhat freely by palaeobiologists (for inference on fossil taxa) and not so often adequately tested before, especially in large datasets comprising species of different phylogenetic lineages. With this regard, this manuscript is a very welcome step forward and will attract the interests of a wide audience of zoologists, evolutionary biologists, palaeobiologists.

I enjoyed reading through the manuscript and I found its content interesting, and the analyses well thought.

I appreciated the cautious approach that the authors have towards some of their findings, and it helps that they clearly state the limitation of their dataset and dietary group allocation.

Nonetheless, the author comes up with important results and clear recommendations for anyone interested in running similar analyses.

I found a small number of issues that I think should be addressed before publication. You can find a list of them below, and others on the attached PDF. Most of them concern the internal organization and structure of the paper rather than its content, so it won't take much to fix it.

MAIN TEXT

While I do not have major issues with the findings, I think that this manuscript will need some modifications before being ready for publication. Most of the changes are small, and would not take much time to address, but I think they would make the manuscript clearer than it currently reads.

My primary concern is that I found somewhat confusing the way the Result section was organised. In particular, the distinction between the Results of the Dorsal-view, Lateral-view GM and linear morphometrics analyses is not particularly clear. I often found myself wondering whether the authors were referring to one or another. Were the dorsal and lateral datasets analysed separately in the GM? And was this distinction maintained also in the Linear measurement? If not, how were the linear measurements compared to either the dorsal or lateral datasets? Some of these details are in the Supplementary Information but should really be clearer in the main text.

How did comparisons between GM and Linear morphometrics worked (if it was quantitatively done)? It should be explicitly specified which datasets were compared. Perhaps, more importantly, is the distinction between the GM and linear morphometrics sections. These should be clearer to the reader. A potential solution could be adding sub-sections in the Method and Result section and discuss them separately.

It is very confusing that the Result section that there is barely mentioned in the linear morphometric dataset. What happened to it?

The Discussion section would also, in my opinion, benefit from being split into 2/3 sub-sections with appropriate headings. It would simply be a matter of moving around some of the paragraphs and expanding some others accordingly. A lot of interesting topics are mentioned, and I think it would strengthen the paper if they were discussed in more detail separately. For instance, I would love to see more group-specific (both extinct and extant) and biomechanics studies be discussed with the results of this manuscript analyses. This is somewhat done briefly in places, but I think that focusing a bit more on the biomechanics would be beneficial for the discussion (e.g. the role of size).

SUPPLEMENTARY INFORMATION

The formatting of references to Figures and Tables and Supplementary Information is not consistent within the main text and between Main text and Supporting Information.

Large parts of the Supplementary Information text (e.g. Statistical analyses) are the exact copy of what is written in the main manuscript. I am not sure why, but some parts do not add anything to what said in the main manuscript. Please carefully revise both documents to avoid repetitions and use the supplementary file for supporting information!

FIGURES

I do like the figures as they are -they are attractive, informative and immediate. I was wondering if the landmarks and measurements figures could be merged with figure 1 of the main text. Landmark and curve numbers should be added to them.

OTHER COMMENTS

Here a series of other comments: I realize that some may be tangential to the purpose of the manuscript, but I thought the authors could be interested in hearing them.

Have you thought to run a Linear discriminant analyses and test how well the PC coordinate of all datasets (Linear morphometrics and GM) predict dietary assignments? In my experience with a variety of Jurassic marine reptiles (ichthyosaurs, plesiosaurs, thalattosuchians) it won't work really well for predicting inferred diet but would assign well taxa to phylogenetic groups (which is perhaps hardly surprising as we use a lot of cranial characters do define different lineages). This is perhaps something that could be added to the Discussion?.

Make sure that you adequately reference all the software that had been used.

A series of minor comments and clarifications are included in the attached PDF.

Finally, I would like to thank the Editor and Authors for the opportunity to review this interesting manuscript and I look forward to seeing it brought to the next stages of publication.

Decision letter (RSOS-202145.R0)

Dear Dr Melstrom

The Editors assigned to your paper RSOS-202145 "The limits of convergence: the roles of phylogeny and dietary ecology in shaping non-avian amniote skulls" have now received comments from reviewers and would like you to revise the paper in accordance with the reviewer comments and any comments from the Editors. Please note this decision does not guarantee eventual acceptance.

Please submit your revised manuscript and required files (see below) no later than 21 days from today's (ie 08-Jun-2021) date. Note: the ScholarOne system will 'lock' if submission of the revision is attempted 21 or more days after the deadline. If you do not think you will be able to meet this deadline please contact the editorial office immediately.

on behalf of Dr Julia Brenda Desojo (Associate Editor) and Kevin Padian (Subject Editor)
openscience@royalsociety.org

Reviewer comments to Author:

Reviewer: 1

Comments to the Author(s)

Reviewer comments

The manuscript is an interesting and creative effort on understanding major patterns in amniote skull morphology. I applaud the creativity and vision of the authors to address the issue and to generate comparable quantitative data from disparate animals. The introductory section is remarkably clear and well written, and so is the discussion.

However, in its present form the manuscript is not yet suitable for publication.

My main concerns have to do with sampling and methodology. Regarding sample, I think the authors can do a better effort to justify the choosing of taxa. I find the exclusion of beaked amniotes poorly elaborated. Clearly, having "heavily modified skulls" could apply also for many toothed amniotes, such as vipers, cetaceans, tapirs, dinosaurs, rodents, dugongs, varanids, etc. In addition, the sample is markedly counterfactual when considering amniote evolution, given is composed of some toothed extant amniotes and excludes not only any fossil taxa but also two major amniote lineages such as birds and turtles. The use of the term "non-avian amniote" is indeed inadequate. In any case, I wonder if an extra emphasis on developing landmarks other than based on teeth could have circumvented this issue and allowed to include some birds and turtles (for example, using the landmark 5 in lateral view to depict also the caudalmost border of the beak, but this is only a suggestion). I do not mean by this to ask the authors to expand the sample and repeat the calculations performed, which are really interesting and have a remarkable heuristic value. Any possible sample will have intrinsic advantages and disadvantages. I only want to stress out that authors should be way more cautious when discussing broad evolutionary implications of their results, given they are representing a quite reduced disparity of cranial morphologies. In several places the authors made assumptions or predictions on how the patterns they found will stand if more lineages were included in the sample, which I find is a bit speculative because authors risk hypothesizing beyond the scope of their results.

Concerning calculations, I consider that lacking of allometry tests is a major flaw of the analysis. This can be viewed even as a missed opportunity of further examination of the data, given that the authors have had all the elements needed to perform allometric regressions, which can be done straightforwardly from the data using the same software and almost without extra computing effort. Justification of not testing influence of size on the shape and ecology is poor and entirely inadequate, in my view. In addition, without testing allometry, all results displayed should be considered "preliminary", so testing for allometry could be considered mandatory in this context. It can be the case, for example, that snout elongation has an allometric component, and this could be being completely overlooked. Insectivory is also constrained by body size. I strongly encourage authors to perform allometric regressions on the sample data, to include them as supplementary information, and to discuss it along the manuscript.

Minor improvements could imply explaining more about the calculations, how they were made, which options were chosen (if Euclidean distances, or Bending Energy, was used for sliding semilandmarks, how models -fits- were constructed for ANOVA or what type of SS was computed, for example). The issue is especially important for the ANOVA and how the models were set, shape~diet, shape*diet, etc. R commands accept many different configurations for each analysis, which have impact on results obtained, therefore clarifying this can be very useful for understanding adequately the methodology. Table 1 appears to be mixing Blomberg's K and phylogenetic ANOVA's results, which render it confusing.

Some of these comments are also explained in the attached pdf file, along with some minor suggestions.

Best regards

Reviewer: 2

Comments to the Author(s)

This is a well-designed study that is very well-presented and written, uses appropriate methodological approaches, and which interprets its results carefully and appropriately in the context of previous work. The authors use geometric morphometrics to explore evolutionary convergence in the skulls of modern mammals, crocodylians and lizards, concluding that phylogeny is critically in defining morphology at the scale of major clades, but that patterns within clades are more complex, and result from an interplay of phylogeny and ecology. I can identify no significant issues with this manuscript and think it could be published without revisions. My only real question was around whether some of the gaps in morphospace between major clades might disappear if fossil taxa were included, and so are perhaps the result of extinction. I am particularly thinking of the much more diverse skull morphologies present in fossil crocodylomorphs, which include short-snouted forms. While I do not think the authors necessarily need to modify their paper, a brief consideration of this point in the discussion might be worthwhile.

Reviewer: 3

Comments to the Author(s)

This is a neat study that provides some interesting and thought-provoking results that will increase our knowledge of phylogenetic constraints and drivers of amniote cranial morphology. I can imagine that this manuscript will be highly cited by morphologists and palaeobiologists, including by myself.

I have uploaded my comments onto the main text PDF using the sticky note tool as I find this much better than writing out all my comments. In most cases my comments and questions are pretty minor and can be easily addressed or answered by the authors.

The only areas of 'concern' that I want to mention regard the time-scaling of the supertree and the phylogenetic analyses employed. I should stress that I do not believe what the authors have done is incorrect, I just have a few questions for more information/clarification and to make a few suggestions about how these analyses could be added to in order for the discussions and interpretations to be made (even) more robust.

1) I'm more familiar with stratigraphic ways of time-calibrating supertrees, so if the below is not 100% relevant I apologise. You mention throughout the importance of identifying phylogenetic signals in your dataset to interpret phylogenetic constraints in skull morphology and convergence. How come you then use only one time-scaling method? Phylogenetic signals can be influenced by branch lengths and inferred divergence dates and therefore it may be prudent to use an additional independent (or two if possible) time-scaling method and test for phylogenetic signal. Examples employed by many palaeobiologists include FDB, hedman, mbl and cal3. With the exception of FDB these time-scaling methods can be easily done in R. If there are similar levels of phylogenetic signal present in a second time-scaled tree then this would show your results are not a false positive from your chosen time-scaling method.

2) You mention in your results: "When alternative phylogenetic hypotheses are tested the results are not significantly impacted, suggesting the broader phylogenetic relationships are driving this pattern" but don't elaborate further. Where are the results for these alternative phylogenies? I understand that you used some different mammal relationships from Upham et al. but did you

use completely different supertrees when testing for alternative phylogenies? You're not incorrect in the way that have resolved polytomies by adding 1 million years to such polytomies and to unresolved divergence dates, but say you had a three way polytomy with taxa A, B and C, how did you decide which taxon of the three diverges first? The best way to account for polytomies and unresolved nodes would be to create 10-20 randomly resolved versions in R (using one of the time-scaling methods mentioned in point 1) and perform the phylosignal analysis (just the K_{mult} or lambda statistic, see point 3) on all of them. If the trees have the same strength of phylogenetic signal then you can proceed with one tree (i.e. the one of you already have) for all other analyses.

If you decide that this way of resolving phylogenetic uncertainty is not 100% relevant/applicable to your manuscript, that is fine, but at the very least please provide more clarification about how many trees you used and the results from them (supp. Info would be fine).

3) The lambda statistic is also good for deducing levels of phylogenetic signal in a dataset (Munkemuller et al. 2012; *Meth. Ecol. Evol.* 3, 743-756). This can be performed using the phylosig package and function of the same name in R. This would provide an independent test of whether PC 1 and PC 2 values for the dorsal and lateral data exhibit phylogenetic signals for the entire dataset.

I want to stress again that I do not think the way you constructed your time-scaled tree is incorrect, I just think because the manuscript hypotheses and analyses hinge on phylogeny, I just want to make sure the methods and outputs are as robust as possible. The only reason I have recommended major revisions is so that you have enough time to explore the additional/alternative methods I recommend. I really want to see this manuscript published and hope that my suggestions can help.

Reviewer: 4

Comments to the Author(s)

Dear Authors and Editor,

This manuscript investigates the links between shape, diet and phylogeny in extant amniote skulls. The link between diet/shape has often been used somewhat freely by palaeobiologists (for inference on fossil taxa) and not so often adequately tested before, especially in large datasets comprising species of different phylogenetic lineages. With this regard, this manuscript is a very welcome step forward and will attract the interests of a wide audience of zoologists, evolutionary biologists, palaeobiologists.

I enjoyed reading through the manuscript and I found its content interesting, and the analyses well thought.

I appreciated the cautious approach that the authors have towards some of their findings, and it helps that they clearly state the limitation of their dataset and dietary group allocation.

Nonetheless, the author comes up with important results and clear recommendations for anyone interested in running similar analyses.

I found a small number of issues that I think should be addressed before publication. You can find a list of them below, and others on the attached PDF. Most of them concern the internal organization and structure of the paper rather than its content, so it won't take much to fix it.

MAIN TEXT

While I do not have major issues with the findings, I think that this manuscript will need some modifications before being ready for publication. Most of the changes are small, and would not take much time to address, but I think they would make the manuscript clearer than it currently reads.

My primary concern is that I found somewhat confusing the way the Result section was organised. In particular, the distinction between the Results of the Dorsal-view, Lateral-view GM and linear morphometrics analyses is not particularly clear. I often found myself wondering whether the authors were referring to one or another. Were the dorsal and lateral datasets analysed separately in the GM? And was this distinction maintained also in the Linear measurement? If not, how were the linear measurements compared to either the dorsal or lateral datasets? Some of these details are in the Supplementary Information but should really be clearer in the main text.

How did comparisons between GM and Linear morphometrics worked (if it was quantitatively done)? It should be explicitly specified which datasets were compared. Perhaps, more importantly, is the distinction between the GM and linear morphometrics sections. These should be clearer to the reader. A potential solution could be adding sub-sections in the Method and Result section and discuss them separately.

It is very confusing that the Result section that there is barely mentioned in the linear morphometric dataset. What happened to it?

The Discussion section would also, in my opinion, benefit from being split into 2/3 sub-sections with appropriate headings. It would simply be a matter of moving around some of the paragraphs and expanding some others accordingly. A lot of interesting topics are mentioned, and I think it would strengthen the paper if they were discussed in more detail separately. For instance, I would love to see more group-specific (both extinct and extant) and biomechanics studies be discussed with the results of this manuscript analyses. This is somewhat done briefly in places, but I think that focusing a bit more on the biomechanics would be beneficial for the discussion (e.g. the role of size).

SUPPLEMENTARY INFORMATION

The formatting of references to Figures and Tables and Supplementary Information is not consistent within the main text and between Main text and Supporting Information. Large parts of the Supplementary Information text (e.g. Statistical analyses) are the exact copy of what is written in the main manuscript. I am not sure why, but some parts do not add anything to what said in the main manuscript. Please carefully revise both documents to avoid repetitions and use the supplementary file for supporting information!

FIGURES

I do like the figures as they are -they are attractive, informative and immediate. I was wondering if the landmarks and measurements figures could be merged with figure 1 of the main text. Landmark and curve numbers should be added to them.

OTHER COMMENTS

Here a series of other comments: I realize that some may be tangential to the purpose of the manuscript, but I thought the authors could be interested in hearing them.

Have you thought to run a Linear discriminant analyses and test how well the PC coordinate of all datasets (Linear morphometrics and GM) predict dietary assignments? In my experience with a variety of Jurassic marine reptiles (ichthyosaurs, plesiosaurs, thalattosuchians) it won't work really well for predicting inferred diet but would assign well taxa to phylogenetic groups (which is perhaps hardly surprising as we use a lot of cranial characters to define different lineages). This is perhaps something that could be added to the Discussion?.

Make sure that you adequately reference all the software that had been used.

A series of minor comments and clarifications are included in the attached PDF.

Finally, I would like to thank the Editor and Authors for the opportunity to review this interesting manuscript and I look forward to seeing it brought to the next stages of publication.

===PREPARING YOUR MANUSCRIPT===

===PREPARING YOUR REVISION IN SCHOLARONE===

<https://royalsociety.org/journals/authors/author-guidelines/#supplementary-material> to include a suitable title and informative caption. An example of appropriate titling and captioning may be found at [https://figshare.com/articles/Table_S2_from_Is_there_a_trade-off_between_peak_performance_and_performance_breadth_across_temperatures_for_aerobic_sc ope_in_teleost_fishes_/3843624](https://figshare.com/articles/Table_S2_from_Is_there_a_trade-off_between_peak_performance_and_performance_breadth_across_temperatures_for_aerobic_scope_in_teleost_fishes_/3843624).

Author's Response to Decision Letter for (RSOS-202145.R0)

See Appendix C.

RSOS-202145.R1 (Revision)

Review form: Reviewer 3

Is the manuscript scientifically sound in its present form?

Yes

Are the interpretations and conclusions justified by the results?

Yes

Is the language acceptable?

Yes

Do you have any ethical concerns with this paper?

No

Have you any concerns about statistical analyses in this paper?

No

Recommendation?

Accept as is

Comments to the Author(s)

I am 100% satisfied with the revisions and changes that have been made to the manuscript. The authors have also presented valid and justified reasons for the reviewer comments they have not taken on board. I'd be happy for this manuscript to be published in its current format.

Review form: Reviewer 4

Is the manuscript scientifically sound in its present form?

Yes

Are the interpretations and conclusions justified by the results?

Yes

Is the language acceptable?

Yes

Do you have any ethical concerns with this paper?

No

Have you any concerns about statistical analyses in this paper?

No

Recommendation?

Accept as is

Comments to the Author(s)

Dear Authors and Editor,

This is the second time I review this manuscript discussing dietary strategies, shape of modern toothed amniotes. In the first version I had only a few comments – mostly minor.

I am very pleased and satisfied that my and my fellow reviewers' comments have been diligently addressed, and that the authors went the "extra mile" in doing so. I particularly appreciated the extra analyses looking into the role of allometry on skull shape as well as the clarifications on sampling and the reorganization of some sections in the text. The first changes address a

limitation of the first version, while the latter modifications are help in the exposition and replicability. Overall, I think that this work represents a very useful (and overdue) resource for researchers interested in the subject.

On request of mine and of another reviewer, a figure has been moved and modified from the supplementary material to the main text, and minor grammatical/ references mistakes have been all corrected.

When the authors decided against implementing the suggestion of a reviewer, they explained the reasons behind their decisions. I found that in all circumstances the reasons provided are valid. I look forward to seeing the future studies will be focusing on the fossil record. 1

All modifications have been implemented and new analyses have considerably improved the quality of the first version of the manuscript. This process has also improved the readability and clarity of the manuscript (that was a weakness of certain areas of the previous version of this work). The figures are informative and eye-catching. The Supplementary file has also been appropriately revised and the repetitions with the main manuscript have been removed.

Overall, I would like to take the time to congratulate the authors for the good work and I look forward to seeing this work in its final published form.

Decision letter (RSOS-202145.R1)

Dear Dr Melstrom,

It is a pleasure to accept your manuscript entitled "The limits of convergence: the roles of phylogeny and dietary ecology in shaping non-avian amniote skulls" in its current form for publication in Royal Society Open Science. The comments of the reviewer(s) who reviewed your manuscript are included at the foot of this letter.

Kind regards,
Royal Society Open Science Editorial Office
Royal Society Open Science

on behalf of Prof Kevin Padian (Subject Editor)
openscience@royalsociety.org

Reviewer comments to Author:

Reviewer: 3

Comments to the Author(s)

I am 100% satisfied with the revisions and changes that have been made to the manuscript. The authors have also presented valid and justified reasons for the reviewer comments they have not taken on board. I'd be happy for this manuscript to be published in its current format.

Reviewer: 4

Comments to the Author(s)

Dear Authors and Editor,

This is the second time I review this manuscript discussing dietary strategies, shape of modern toothed amniotes. In the first version I had only a few comments – mostly minor.

I am very pleased and satisfied that my and my fellow reviewers' comments have been diligently addressed, and that the authors went the "extra mile" in doing so. I particularly appreciated the extra analyses looking into the role of allometry on skull shape as well as the clarifications on sampling and the reorganization of some sections in the text. The first changes address a limitation of the first version, while the latter modifications are help in the exposition and replicability. Overall, I think that this work represents a very useful (and overdue) resource for researchers interested in the subject.

On request of mine and of another reviewer, a figure has been moved and modified from the supplementary material to the main text, and minor grammatical/references mistakes have been all corrected.

When the authors decided against implementing the suggestion of a reviewer, they explained the reasons behind their decisions. I found that in all circumstances the reasons provided are valid. I look forward to seeing the future studies will be focusing on the fossil record. 1

All modifications have been implemented and new analyses have considerably improved the quality of the first version of the manuscript. This process has also improved the readability and clarity of the manuscript (that was a weakness of certain areas of the previous version of this work). The figures are informative and eye-catching. The Supplementary file has also been appropriately revised and the repetitions with the main manuscript have been removed.

Overall, I would like to take the time to congratulate the authors for the good work and I look forward to seeing this work in its final published form.

Appendix A**ROYAL SOCIETY
OPEN SCIENCE****The limits of convergence: the roles of phylogeny and dietary ecology in shaping non-avian amniote skulls**

Journal:	Royal Society Open Science
Manuscript ID	RSOS-202145
Article Type:	Research
Date Submitted by the Author:	30-Nov-2020
Complete List of Authors:	Melstrom, Keegan; Natural History Museum of Los Angeles County, Dinosaur Institute; University of Utah, Department of Geology and Geophysics; Natural History Museum of Utah Angielczyk, Kenneth; The Field Museum, Department of Geology Ritterbush, Kathleen; University of Utah, Geology and Geophysics Irmis, Randall; University of Utah, Geology and Geophysics; Natural History Museum of Utah
Subject:	ecology < BIOLOGY, evolution < BIOLOGY
Keywords:	Amniotes, Convergence, Cranial Morphology, Diet, Macroevolution
Subject Category:	Organismal and Evolutionary Biology

Author-supplied statements

Relevant information will appear here if provided.

Ethics

Does your article include research that required ethical approval or permits?:

This article does not present research with ethical considerations

Statement (if applicable):

CUST_IF_YES_ETHICS :No data available.

Data

It is a condition of publication that data, code and materials supporting your paper are made publicly available. Does your paper present new data?:

Yes

Statement (if applicable):

Linear morphometric data are available in supplemental information and photographs are available in the Dryad Digital Repository: <https://doi.org/10.5061/dryad.0k6djh9xc>.

<https://datadryad.org/stash/share/OnoSWGKGSZnc3HEaTb8Av49ZvRPe4Qhtf8yZqcl0hdY> (will start download of photo dataset)

Conflict of interest

I/We declare we have no competing interests

Statement (if applicable):

CUST_STATE_CONFLICT :No data available.

Authors' contributions

This paper has multiple authors and our individual contributions were as below

Statement (if applicable):

K.M.M. and K.D.A. collected the data. K.M.M., R.B.I., and K.A.R. conceived the study and designed the analyses. All authors prepared the manuscript.

The limits of convergence: the roles of phylogeny and dietary ecology in shaping non-avian amniote skulls

Keegan M. Melstrom^{a,b,c*}, Kenneth D. Angielczyk^d, Kathleen A. Ritterbush^b, and Randall B. Irmis^{b,c}

^a *Dinosaur Institute, Natural History Museum of Los Angeles County, 900 W Exposition Blvd, Los Angeles, CA 90007*; ^b *Department of Geology and Geophysics, University of Utah, 115 S 1460 E, Salt Lake City, UT 84112-0102, USA*; ^c *Natural History Museum of Utah, University of Utah, 301 Wakara Way, Salt Lake City, UT 84108-1214, USA*; ^d *Field Museum of Natural History, 1400 South Lake Shore Drive, Chicago, IL 60605-2496, USA.*

Keywords: Amniotes, Convergence, Cranial Morphology, Diet, Macroevolution

1. Summary

Cranial morphology is remarkably varied in living amniotes, ranging from short-faced mammals to the elongate snouts of crocodylians. This diversity of shapes is thought to correspond with feeding ecology, a relationship repeatedly demonstrated at smaller phylogenetic scales, but one that remains untested across amniote phylogeny. Using a combination of 2D geometric and linear morphometrics, we investigate the links between phylogenetic relationships, diet, and skull shape in an expansive dataset of extant amniotes with teeth: mammals, lepidosaurs, and crocodylians. We find that both phylogeny and diet have statistically significant effects on skull shape, although these effects differ depending on the dataset analyzed. The three major clades largely partition morphospace, each plotting in separate regions with limited overlap. Mammals and squamates extensively diversify within their respective regions. Among all three groups, dietary generalists often occupy clade-specific central regions of morphospace. Some parallel changes in skull shape occur in clades with distinct evolutionary histories but similar diets. However, members of a given clade often present distinct skull shape solutions for a given diet, and the vast majority of species retain the unique aspects of their ancestral skull plan, underscoring the limits of morphological convergence due to ecology in amniotes. These data demonstrate that certain skull shapes may provide functional advantages suited to particular diets, but accounting for both phylogenetic history and ecology can provide a more nuanced approach for inferring the ecology and functional morphology of cryptic or extinct amniotes.

2. Introduction

[revised manuscript text omitted]

Some major groups, however, are not represented in this study, primarily due to the absence of homologous points or key landmarks not being visible in either dorsal or lateral view. In particular, primates and cetaceans are excluded from the mammalian dataset. These groups are characterized by extremes in morphology, especially cetaceans due to their extraordinary size and the dramatic modifications associated with a pelagic lifestyle. Previous research has compared the skull shape of crocodylians and toothed whales but used functionally analogous landmarks as opposed to the homologous landmarks that we use here [23]. Similarly, the morphological extremes of primates (e.g. tall skulls, forward-facing orbits, etc.) obscure the majority of dorsal landmarks used in this study. We did not sample birds and turtles because both groups are edentulous and **have heavily modified skulls**. In the case of squamates, we did not sample snakes. This group makes up a large proportion of squamate diversity and displays remarkable cranial disparity, including the modification of many sutures (e.g. premaxilla-maxilla) and reduction in maxillary tooth row, precluding the use of a number of our homologous landmarks. Additionally, some snake taxa no longer possess homologous semilandmark curves in dorsal and lateral views.

3.2 Morphological data

We quantified cranial morphologies using a combination of linear and 2D GM (Fig. 2). We photographed skulls using a standardized procedure and limited set of camera lenses to reduce measurement error [34,35]. For the dorsal dataset, we digitized ten landmarks and 35 sliding semilandmarks, which together capture the broad outline of half of the skull and key cranial features such as preorbital length, size and location of the orbit, and general morphology of the back of the skull (electronic supplementary material). Four sliding semilandmark curves separated by landmarks 1–5 collectively outlined the lateral skull margin. In left lateral view, seven landmarks highlight key characteristics such as snout length, orbit location, orbit size, length of tooth row, and location of important jaw muscle attachments. Owing to an absence of repeatable, homologous landmarks on ventral or dorsal margins in lateral view, 40 sliding semilandmarks broken into three curves quantify the morphology of these regions. Although these methods capture less phenotypic information than 3D GM analyses, well-designed 2D GM schemes detect major morphological patterns [21,36–38]. Importantly, 3D models generated from surface scans or CT data are sensitive to distortion, which can limit any future comparisons to fossils specimens.

We used the multipoint tool of **Fiji** to digitize landmarks and the Bezier Curve tool to digitize semilandmark curves [39]. We then separately saved landmarks and semilandmark curves as .txt files, which were subsequently processed in **RStudio** (version 1.2.5033; R version 3.6.2). Following a Generalized **Procrustes Analysis**, which removes variation in location, orientation, and scale, we performed a principal components analysis on the Procrustes coordinates to summarize shape variation and produce a set of uncorrelated shape variables [40–43].

We also investigated shape variation using linear morphometrics and collected 10 measurements using digital calipers (Neiko Tools) and a tailor's measuring tape for curved surfaces (e.g. length of tooth row) or measurements greater than 200 mm. We measured specimens directly (i.e. not from photographs) to avoid **errors associated with sample orientation and variation in scale bar position relative to measurements**. **To ensure size did not bias the sample**, all measurements were normalized against total skull length. Although size does play a key role in ecology, such as in influencing prey acquisition in predators, **this study focuses on shape and the removal of size also facilitates more direct comparisons with the GM dataset**.

3.3 Phylogenetic hypotheses

Given the broad taxonomic range of this study, we generated a time-calibrated phylogenetic tree built from a combination of sources (Fig. 2). The topology for Squamata was based on the molecular dataset of Pyron et al. [44], whereas the tree for Crocodylia was based on Erickson et al.'s [45] reanalysis of molecular

data from Gatsey et al. [46]. The mammal topology utilized a variety of sources for individual groups
including carnivorans, marsupials, and bats [43–48]. We then built a nexus file in Mesquite [50], in which these
relatively well-agreed on topologies were combined. Timetree.org, which synthesizes divergence dates
derived from molecular data, was used to estimate divergence dates for calculating branch lengths [51].
Subsequent statistical analyses failed in cases of polytomies or unresolved nodes, so we added 1 million years
to a divergence date in these situations. To test the effect of phylogenetic uncertainty, we modified our
mammal trees using alternative hypotheses generated primarily from Upham et al. [52].

**3.4 Ecological categories**

Amniotes exhibit a remarkable range in diets, from extreme specialists (e.g. anteaters in both
mammals and squamates) to broad generalists. We placed sampled taxa into four coarse dietary categories:
carnivore, insectivore, omnivore, and herbivore (electronic supplementary material). Categorizations were
obtained from the literature, including large databases and monographic sources (e.g. refs. 3,49–53 and
references therein). Although studies with a narrower taxonomic focus often have more precise dietary
categories (e.g. terrestrial invertebrates, aquatic invertebrates, terrestrial vertebrates, fish, carrion, fruit, seeds,
nectar, and other plant material; 11,38,54–56), we used broader dietary groupings to better facilitate
comparisons between distantly related groups. This categorization also improves consistency for taxa whose
diets are more poorly known, which is the case for many squamates.

Following previous studies investigating the relationship between skull shape and diet, we assigned
taxa to a dietary category if they obtain the majority of food from that source [13,14,23,30]. Carnivores are
defined as organisms that consume vertebrate material, or large invertebrates (e.g. squid) in the case of marine
taxa, for approximately 90% of their diet. Insectivores are categorized as animals that primarily (~90%) eat
terrestrial invertebrates. Many taxa have a diverse array of food sources, ranging from insects to plant
material. In these cases, we categorize the animal as an omnivore. We designate specimens as herbivores if
they consume plant material, such as algae, fruit, and grass, for approximately 90% of their food. This
categorization method can accommodate herbivores that periodically consume a minor component of animal
material (either vertebrate or invertebrate), as well as low- and high-fiber herbivores. Ultimately, diet is
flexible and comprises a multivariate set of continuous variables, so the selection of a cut-off of roughly 10%
plant or animal matter in specialist categories (i.e. carnivore and herbivore) is arbitrary, but it allows for taxa
that periodically consume novel material to remain in their main dietary group.

**3.5 Statistical analyses**

To test for covariation between taxonomic and ecological groups and morphology, we utilized a
combination of statistical tests on linear and geometric morphometric data. To evaluate the amount of shape
variation attributable to diet and phylogeny in geometric morphometric analyses, we performed a Procrustes
ANOVA using the `procD.lm` function from the ‘geomorph’ R package. This function quantifies the amount of
morphological variation that can be attributed to one or more factors and is appropriate for use with high-
dimensional data (e.g. 10,11,57). A Procrustes ANOVA does not take phylogenetic relationships into account
and phylogeny was evaluated by categorizing each specimen into one of three groups: Mammalia, Crocodylia,
and Lepidosauria (*Supplementary Table 1*). Similarly, diet was assigned to one of four categories: carnivore,
insectivore, omnivore, and herbivore (Section 2.4, *Supplementary Table 1*). For the second analysis, we used the
`procD.pgls` function in ‘geomorph’, which performs a Procrustes ANOVA in a phylogenetic framework with
9999 iterations to evaluate the effect of dietary ecology on skull shape on the GM datasets [61]. This analysis
tests for the shape variation attributable to diet with time-calibrated phylogenetic relationships explicitly
taken into account. Taxa that are closely related will tend to share similar traits. In particular, cranial
morphology has previously been shown to contain a significant phylogenetic signal (e.g. [8,28,29,62,63]). We
evaluated the degree of phylogenetic signal in shape variables relative to what would be expected by
Brownian motion by calculating K_{mult} , a multivariate generalization of the K statistic [61].

To evaluate the linear morphometric data, we first had to combine continuous linear data with
discrete ecological categories (section 2.4) Using the Gower distance metric, a coefficient used to measure the
similarity between different samples [40], we separately transformed linear measurements and ecological
categories allowing us to compare the similarities between mixed categorical (i.e. diet in this study) and
continuous data (i.e. morphology). Following this, we performed a principal coordinate analysis (PCO) on the
dissimilarity matrix to generate a dataset of continuous variables [58,64]. We used a canonical correlation
analysis (CCA) to plot the relationship between morphological and ecological PCO scores (described in 2.4).
CCA finds linear correlations between two datasets that allow one dataset to maximally predict the second set.

In this case, the CCA produces a dataset that allows morphology to maximally predict dietary ecology. Finally, for the linear morphometric dataset, we used the aov function from the 'stats' package to perform an ANOVA to investigate for differences in mean shape across the major phylogenetic groups generated from the CCA data.

4. Results

In both GM datasets, the first three principal component (PC) axes summarize over 75% of skull variation, with PC1 accounting for approximately 50% in both cases (Figs. 3 and 4). PC1 primarily describes variation associated with snout dimensions, with rostrum width and height playing key roles in the dorsal and lateral datasets, respectively. Specimens with exceptionally short snouts plot on one extreme (lowest PC1 scores) and are primarily carnivorous mammalian mammals, whereas the other morphological extreme (elongate, narrow snouts) is exemplified by *Tomistoma* and *Gavialis*, two carnivorous crocodylians (Fig. 1). PC2 describes different variation in the dorsal and lateral datasets. In the former, PC2 primarily summarizes the shape associated with the anterior end of the snout and orbital morphology. At low PC2 scores, the region near the premaxilla is narrow and rounded whereas the orbits are relatively small. In contrast, orbits are large and the anterior snout is square and relatively wide for high PC2 scores. In lateral view, PC2 primarily describes variation in length of the dorsal and ventral skull margins, with minor input from the relative height of the orbit. In the dorsal dataset PC3 describes variation in anterior snout shape, the lateral margin flanking the orbits, and the location of the posterior portion of the parietal, whereas the lateral dataset PC3 describes variation in the length of the orbits and the height of the skull. PC4 and higher axes each account for less than 5% of variance.

Major clades (i.e. mammals, lepidosaurs, and crocodylians) largely occupy separate regions of morphospace in the dorsal and lateral GM analyses (Figs. 3 and 4), with the dorsal dataset best discriminating clades. Only isolated taxa exemplifying morphological extremes (e.g. *Dasypus novemcinctus*, nine-banded armadillo; highest PC 1 score for mammals) plot as outliers of their clade. Mammal and lepidosaur morphospace adjoin in the short-snouted, large orbit region of morphospace, with the carnivorous phocids *Mirounga*, *Erignathus*, and *Halichoerus* exemplifying this extreme in mammals. In the lateral dataset, crocodylians plot away from the other two major clades, whereas the morphospace of lepidosaurs and mammals overlap to a greater degree than the dataset of skulls in dorsal view, although their ranges remain significantly different (Fig. 4; Procrustes ANOVA $p < 0.001$). Aquatic mammals again display extremes in mammalian skull shape, with select phocids and herbivorous sirenians plotting between a diverse selection of squamates (Fig. 4). Morphological overlap is greatest in the linear morphometric dataset, although taxonomy remains a significant influence (ANOVA $p < 0.001$; electronic supplementary material). Similar to GM datasets, long-snouted crocodylians plot by themselves, but short-snouted taxa (*Paleosuchus* and *Osteolaemus*) overlap with mammals. This overlap also extends to mammals and lepidosaurs and is more pronounced than in either lateral or dorsal GM datasets.

Mammals display a significantly greater amount of morphological disparity (measured as sum of variances) than lepidosaurs or crocodylians in GM analyses (Welch Two Sample t-test $p < 0.001$), in part driven by the remarkably diverse skulls of marine and flying taxa (Figs. 3 and 4, electronic supplementary material). Lepidosaurs have a lower degree of morphological disparity than mammals. In contrast, despite their low species richness compared to mammals and squamates, crocodylians occupy a relatively large range of morphospace, although it is still the lowest of tested clades (electronic supplementary material). It is worth noting that these results are an underestimate of total disparity, especially for mammals and lepidosaurs, as we did not include many groups that would certainly exhibit significant shape diversity, such as cetaceans, primates, and many groups of limbless lizards. **In spite of this, we think that these results reflect broad disparity patterns in living amniotes, as we sought to sample taxa that express much of the morphological variation exhibited by their clades.**

When phylogeny is accounted for, the relationship between diet and skull shape in the tested GM datasets reveals conflicting assessments (Table 1). In dorsal view, this relationship is not statistically significant ($p = 0.088$), whereas in lateral view it is significant ($p = 0.006$). In both cases, the goodness of fit is low ($R^2 = 0.029$ and 0.044 , respectively), which demonstrates that the predictive power of this relationship is weak. The tests of phylogenetic signal indicate that both dorsal and lateral shapes have less signal than

1 expected under a Brownian motion model of evolution ($K_{\text{mult}} = 0.331$ and 0.394 for dorsal and lateral datasets,
2 respectively) and **that variance tends to be distributed more within clades than between clades**. When
3 alternative phylogenetic hypotheses are tested the results are not significantly impacted, suggesting the
4 broader phylogenetic relationships are driving this pattern.

[revised manuscript text omitted]

of dental elements [23,30]. These data, as well as the linear analyses here, contain less morphological information, which may help facilitate the overlap found in those studies. Furthermore, the morphological differences driving the separations we observed are not present in unsampled taxa: no marine mammal has enlarged pterygoids comparable to those of the crocodylians in the dataset, regardless of snout length. For these reasons, we predict that these unoccupied regions reflect reality rather than unsampled taxa. Understanding the constraints preventing greater overlap may be a fruitful pursuit for future studies because they may offer insights into the morphological limits of each clade.

Diet has a significant influence on skull shape in the sampled non-avian amniotes in the lateral dataset, but not in the dorsal dataset. Additionally, we do not find an overarching significant predictive relationship between these variables that universally applies across all three clades. Instead, members of the three major clades evolve a variety of skull shapes associated with a given diet that reflects their idiosyncratic combinations of skull architecture, dental morphology, and degree of emphasis on oral processing of food, corroborating findings for a large sample of avians [14]. This result underscores the importance of phylogenetic scale in form-function investigations. Investigations of individual specific clades often demonstrate more distinct relationships with dietary ecology playing a key role in shape [10,12,67,68]. In contrast, when a wider phylogenetic focus is applied, such as in this study, **phylogeny tends to overprint trophic patterns**, although ecology may continue to play a significant, but less predictive role [5,8,13,14,28,29]. In spite of this result, some degree of convergence is possible, but taking a broad phylogenetic perspective elucidates how general functional principles and specific dietary and other demands map onto particular skull architectures to produce similar or different skull shapes [69].

[revised manuscript text omitted]

Tables

Inserted at the end of the document.

Figures

Figure 1

Figure 2

Figure 3

Figure 4

Figure and table captions

Table 1. Summary statistics of Procrustes ANOVA and phylogenetic ANOVAs testing the relationship between diet, phylogenetic relationships (Procrustes ANOVA only), and skull shape using geometric morphometric shape data on dorsal and lateral datasets. Regressions were run using ‘geomorph’ package.

Figure 1. Representative non-avian amniote skulls used in this study scaled to the same skull length to emphasize proportional differences. A) *Ailurus fulgens* (FMNH 36755) B) *Galerella sanguinea* (FMNH 83649), C) *Sphenodon punctatus* (UF 11978), D) *Dendrolagus lumholtzi* (FMNH 60897); E) *Varanus salvator* (UF 56879); F) *Paleosuchus palpebrosus* (FMNH 69869); H) *Crocodylus niloticus* (TMM M-1786); I) *Gavialis gangeticus* (NHM R. Unnumbered). Abbreviations: FMNH, Field Museum of Natural History, Chicago; NHM, Natural History Museum, London; TMM, Texas Vertebrate Paleontology Collections at The University of Texas, Austin; UF, University of Florida, University of Florida, Gainesville. Scale bar equals 3 cm.

Figure 2. Time calibrated phylogenetic tree of studied non-avian amniotes. Major clades are noted whereas diet is illustrated by terminal branch color. Gray circles represent 100 million years. Silhouettes courtesy phylopic.org and Sarah Werning.

Figure 3. Plot of principle components 1 and 2 for the dorsal geometric morphometric dataset. Clades are represented by shapes and diets with colors. The three major clades plot in separate regions of morphospace with relatively little overlap. Convex hulls outline extent of each clade’s morphospace. Extremes of skull shape along each axis shown using landmarks (anterior to left).

Figure 4. Plot of principle components 1 and 2 for the lateral geometric morphometric dataset. Clades are represented by shapes and diets with colors. Similar to the dorsal dataset, crocodylians plot in separate regions of morphospace, although lepidosaurs and mammals overlap to a greater degree. Convex hulls

outline extent of each clade's morphospace. Extremes of skull shape along each axis shown using landmarks (anterior to left).

Table 1

		Dorsal Dataset							
		DF	SS	MS	R ²	F	Z	P	K _{mult}
Procrustes ANOVA	Diet	3	0.320	0.107	0.069	3.737	3.197	<0.001	
	Residuals	152	4.349	0.029	0.931				
	Total	155	4.669						
	Taxonomy	2	2.131	1.065	0.456	64.183	8.124	<0.001	
	Residuals	153	2.539	0.017	0.544				
	Total	155	4.690						
Phylogenetic ANOVA	Diet	3	0.002	0.0006	0.029	1.564	1.379	0.0882	0.331
	Residuals	152	0.058	0.0004	0.970				
	Total	155	0.060						
		Lateral Dataset							
Procrustes ANOVA	Diet	3	0.300	0.100	0.042	2.129	1.806	0.037	
	Residuals	145	6.813	0.047	0.958				
	Total	148	7.113						
	Taxonomy	2	2.788	1.394	0.392	47.038	7.014	<0.001	
	Residuals	146	4.326	0.030	0.608				
	Total	148	7.114						
Phylogenetic ANOVA	Diet	3	0.003	0.001	0.044	2.207	2.484	0.006	0.394
	Residuals	145	0.075	0.001	0.956				
	Total	148	0.078						

Appendix B**ROYAL SOCIETY
OPEN SCIENCE****The limits of convergence: the roles of phylogeny and dietary ecology in shaping non-avian amniote skulls**

Journal:	Royal Society Open Science
Manuscript ID	RSOS-202145
Article Type:	Research
Date Submitted by the Author:	30-Nov-2020
Complete List of Authors:	Melstrom, Keegan; Natural History Museum of Los Angeles County, Dinosaur Institute; University of Utah, Department of Geology and Geophysics; Natural History Museum of Utah Angielczyk, Kenneth; The Field Museum, Department of Geology Ritterbush, Kathleen; University of Utah, Geology and Geophysics Irmis, Randall; University of Utah, Geology and Geophysics; Natural History Museum of Utah
Subject:	ecology < BIOLOGY, evolution < BIOLOGY
Keywords:	Amniotes, Convergence, Cranial Morphology, Diet, Macroevolution
Subject Category:	Organismal and Evolutionary Biology

Author-supplied statements

Relevant information will appear here if provided.

Ethics

Does your article include research that required ethical approval or permits?:

This article does not present research with ethical considerations

Statement (if applicable):

CUST_IF_YES_ETHICS :No data available.

Data

It is a condition of publication that data, code and materials supporting your paper are made publicly available. Does your paper present new data?:

Yes

Statement (if applicable):

Linear morphometric data are available in supplemental information and photographs are available in the Dryad Digital Repository: <https://doi.org/10.5061/dryad.0k6djh9xc>.

<https://datadryad.org/stash/share/OnoSWGKGSZnc3HEaTb8Av49ZvRPe4Qhtf8yZqcl0hdY> (will start download of photo dataset)

Conflict of interest

I/We declare we have no competing interests

Statement (if applicable):

CUST_STATE_CONFLICT :No data available.

Authors' contributions

This paper has multiple authors and our individual contributions were as below

Statement (if applicable):

K.M.M. and K.D.A. collected the data. K.M.M., R.B.I., and K.A.R. conceived the study and designed the analyses. All authors prepared the manuscript.

The limits of convergence: the roles of phylogeny and dietary ecology in shaping non-avian amniote skulls

Keegan M. Melstrom^{a,b,c*}, Kenneth D. Angielczyk^d, Kathleen A. Ritterbush^b, and Randall B. Irmis^{b,c}

^a Dinosaur Institute, Natural History Museum of Los Angeles County, 900 W Exposition Blvd, Los Angeles, CA 90007; ^b Department of Geology and Geophysics, University of Utah, 115 S 1460 E, Salt Lake City, UT 84112-0102, USA; ^c Natural History Museum of Utah, University of Utah, 301 Wakara Way, Salt Lake City, UT 84108-1214, USA; ^d Field Museum of Natural History, 1400 South Lake Shore Drive, Chicago, IL 60605-2496, USA.

Keywords: Amniotes, Convergence, Cranial Morphology, Diet, Macroevolution

1. Summary

Cranial morphology is remarkably varied in living amniotes, ranging from short-faced mammals to the elongate snouts of crocodylians. This diversity of shapes is thought to correspond with feeding ecology, a relationship repeatedly demonstrated at smaller phylogenetic scales, but one that remains untested across amniote phylogeny. Using a combination of 2D geometric and linear morphometrics, we investigate the links between phylogenetic relationships, diet, and skull shape in an expansive dataset of extant amniotes with teeth: mammals, lepidosaurs, and crocodylians. We find that both phylogeny and diet have statistically significant effects on skull shape, although these effects differ depending on the dataset analyzed. The three major clades largely partition morphospace, each plotting in separate regions with limited overlap. Mammals and squamates extensively diversify within their respective regions. Among all three groups, dietary generalists often occupy clade-specific central regions of morphospace. Some parallel changes in skull shape occur in clades with distinct evolutionary histories but similar diets. However, members of a given clade often present distinct skull shape solutions for a given diet, and the vast majority of species retain the unique aspects of their ancestral skull plan, underscoring the limits of morphological convergence due to ecology in amniotes. These data demonstrate that certain skull shapes may provide functional advantages suited to particular diets, but accounting for both phylogenetic history and ecology can provide a more nuanced approach for inferring the ecology and functional morphology of cryptic or extinct amniotes.

2. Introduction

[revised manuscript text omitted]

Some major groups, however, are not represented in this study, primarily due to the absence of homologous points or key landmarks not being visible in either dorsal or lateral view. In particular, primates and cetaceans are excluded from the mammalian dataset. These groups are characterized by extremes in morphology, especially cetaceans due to their extraordinary size and the dramatic modifications associated with a pelagic lifestyle. Previous research has compared the skull shape of crocodylians and toothed whales but used functionally analogous landmarks as opposed to the homologous landmarks that we use here [23]. Similarly, the morphological extremes of primates (e.g. tall skulls, forward-facing orbits, etc.) obscure the majority of dorsal landmarks used in this study. We did not sample birds and turtles because both groups are edentulous and have heavily modified skulls. In the case of squamates, we did not sample snakes. This group makes up a large proportion of squamate diversity and displays remarkable cranial disparity, including the modification of many sutures (e.g. premaxilla-maxilla) and reduction in maxillary tooth row, precluding the use of a number of our homologous landmarks. Additionally, some snake taxa no longer possess homologous semilandmark curves in dorsal and lateral views.

3.2 Morphological data

We quantified cranial morphologies using a combination of linear and 2D GM (Fig. 2). We photographed skulls using a standardized procedure and limited set of camera lenses to reduce measurement error [34,35]. For the dorsal dataset, we digitized ten landmarks and 35 sliding semilandmarks, which together capture the broad outline of half of the skull and key cranial features such as preorbital length, size and location of the orbit, and general morphology of the back of the skull (electronic supplementary material). Four sliding semilandmark curves separated by landmarks 1–5 collectively outlined the lateral skull margin. In left lateral view, seven landmarks highlight key characteristics such as snout length, orbit location, orbit size, length of tooth row, and location of important jaw muscle attachments. Owing to an absence of repeatable, homologous landmarks on ventral or dorsal margins in lateral view, 40 sliding semilandmarks broken into three curves quantify the morphology of these regions. Although these methods capture less phenotypic information than 3D GM analyses, well-designed 2D GM schemes detect major morphological patterns [21,36–38]. Importantly, 3D models generated from surface scans or CT data are sensitive to distortion, which can limit any future comparisons to fossils specimens.

We used the multipoint tool of FIJI to digitize landmarks and the Bezier Curve tool to digitize semilandmark curves [39]. We then separately saved landmarks and semilandmark curves as .txt files, which were subsequently processed in RStudio (version 1.2.5033; R version 3.6.2). Following a Generalized Procrustes Analysis, which removes variation in location, orientation, and scale, we performed a principal components analysis on the Procrustes coordinates to summarize shape variation and produce a set of uncorrelated shape variables [40–43].

We also investigated shape variation using linear morphometrics and collected 10 measurements using digital calipers (Neiko Tools) and a tailor's measuring tape for curved surfaces (e.g. length of tooth row) or measurements greater than 200 mm. We measured specimens directly (i.e. not from photographs) to avoid errors associated with sample orientation and variation in scale bar position relative to measurements. To ensure size did not bias the sample, all measurements were normalized against total skull length. Although size does play a key role in ecology, such as in influencing prey acquisition in predators, this study focuses on shape and the removal of size also facilitates more direct comparisons with the GM dataset.

3.3 Phylogenetic hypotheses

Given the broad taxonomic range of this study, we generated a time-calibrated phylogenetic tree built from a combination of sources (Fig. 2). The topology for Squamata was based on the molecular dataset of Pyron et al. [44], whereas the tree for Crocodylia was based on Erickson et al.'s [45] reanalysis of molecular

data from Gatsey et al. [46]. The mammal topology utilized a variety of sources for individual groups
including carnivorans, marsupials, and bats [43–48]. We then built a nexus file in Mesquite [50], in which these
relatively well-agreed on topologies were combined. Timetree.org, which synthesizes divergence dates
derived from molecular data, was used to estimate divergence dates for calculating branch lengths [51].
Subsequent statistical analyses failed in cases of polytomies or unresolved nodes, so we added 1 million years
to a divergence date in these situations. To test the effect of phylogenetic uncertainty, we modified our
mammal trees using alternative hypotheses generated primarily from Upham et al. [52].

3.4 Ecological categories

Amniotes exhibit a remarkable range in diets, from extreme specialists (e.g. an ers in both
mammals and squamates) to broad generalists. We pl  sampled taxa into four coarse dietary categories:
carnivore, insectivore, omnivore, and herbivore (electronic supplementary material). Categorizations were
obtained from the literature, including large databases and monographic sources (e.g. refs. 3,49–53 and
references therein). Although studies with a narrower taxonomic focus often have more precise dietary
categories (e.g. terrestrial invertebrates, aquatic invertebrates, terrestrial vertebrates, fish, carrion, fruit, seeds,
nectar, and other plant material; 11,38,54–56), we used broader dietary groupings to better facilitate
comparisons between distantly related groups. This categorization also improves consistency for taxa whose
diets are more poorly known, which is the case for many squamates.

Following previous studies investigating the relationship between skull shape and diet, we assigned
taxa to a dietary category if they obtain the majority of food from that source [13,14,23,30]. Carnivores are
defined as organisms that consume vertebrate material, or large invertebrates (e.g. squid) in the case of marine
taxa, for approximately 90% of their diet. Insectivores are categorized as animals that primarily (~90%) eat
terrestrial invertebrates. Many taxa have a diverse array of food sources, ranging from insects to plant
material. In these cases, we categorize the animal as an omnivore. We designate specimens as herbivores if
they consume plant material, such as algae, fruit, and grass, for approximately 90% of their food. This
categorization method can accommodate herbivores that periodically consume a minor component of animal
material (either vertebrate or invertebrate), as well as low- and high-fiber herbivores. Ultimately, diet is
flexible and comprises a multivariate set of continuous variables, so the selection of a cut-off of roughly 10%
plant or animal matter in specialist categories (i.e. carnivore and herbivore) is arbitrary, but it allows for taxa
that periodically consume novel material to remain in their main dietary group.

3.5 Statistical analyses

To test for covariation between taxonomic and ecological groups and morphology, we utilized a
combination of statistical tests on linear and geometric morphometric data. To evaluate the amount of shape
variation attributable to diet and phylogeny in geometric morphometric analyses, we performed a Procrustes
ANOVA using the `procD.lm` function from the ‘geomorph’ R package. This function quantifies the amount of
morphological variation that can be attributed to one or more factors and is appropriate for use with high-
dimensional data (e.g. 10,11,57). A Procrustes ANOVA does not take phylogenetic relationships into account
and phylogeny was evaluated by categorizing each specimen into one of three groups: Mammalia, Crocodylia,
and Lepidosauria (*Supplementary Table 1*). Similarly, diet was assigned to one of four categories: carnivore,
insectivore, omnivore, and herbivore (Section 2.4, *Supplementary Table 1*). For the second analysis, we used the
`procD.pgls` function in ‘geomorph’, which performs a Procrustes ANOVA in a phylogenetic framework with
9999 iterations to evaluate the effect of dietary ecology on skull shape on the GM datasets [61]. This analysis
tests for the shape variation attributable to diet with time-calibrated phylogenetic relationships explicitly
taken into account. Taxa that are closely related will tend to share similar traits. In particular, cranial
morphology has previously been shown to contain a significant phylogenetic signal (e.g. [8,28,29,62,63]). We
evaluated the degree of phylogenetic signal in shape variables relative to what would be expected by
Brownian motion by calculating K_{mult} , a multivariate generalization of the K statistic [61].

To evaluate the linear morphometric data, we first had to combine continuous linear data with
discrete ecological categories (section 2.4) Using the Gower distance metric, a coefficient used to measure the
similarity between different samples [40], we separately transformed linear measurements and ecological
categories allowing us to compare the similarities between mixed categorical (i.e. diet in this study) and
continuous data (i.e. morphology). Following this, we performed a principal coordinate analysis (PCO) on the
dissimilarity matrix to generate a dataset of continuous variables [58,64]. We used a canonical correlation
analysis (CCA) to plot the relationship between morphological and ecological PCO scores (described in 2.4).
CCA finds linear correlations between two datasets that allow one dataset to maximally predict the second set.

In this case, the CCA produces a dataset that allows morphology to maximally predict dietary ecology. Finally, for the linear morphometric dataset, we used the aov function from the 'stats' package to perform an ANOVA to investigate for differences in mean shape across the major phylogenetic groups generated from the CCA data.

4. Results

In both GM datasets, the first three principal component (PC) axes summarize over 75% of skull variation, with PC1 accounting for approximately 50% in both cases (Figs. 3 and 4). PC1 primarily describes variation associated with snout dimensions, with rostrum width and height playing key roles in the dorsal and lateral datasets, respectively. Specimens with exceptionally short snouts plot on one extreme (lowest PC1 scores) and are primarily carnivorous mammalian mammals, whereas the other morphological extreme (elongate, narrow snouts) is exemplified by *Tomistoma* and *Gavialis*, two carnivorous crocodylians (Fig. 1). PC2 describes different variation in the dorsal and lateral datasets. In the former, PC2 primarily summarizes the shape associated with the anterior end of the snout and orbital morphology. At low PC2 scores, the region near the premaxilla is narrow and rounded whereas the orbits are relatively small. In contrast, orbits are large and the anterior snout is square and relatively wide for high PC2 scores. In lateral view, PC2 primarily describes variation in length of the dorsal and ventral skull margins, with minor input from the relative height of the orbit. In the dorsal dataset PC3 describes variation in anterior snout shape, the lateral margin flanking the orbits, and the location of the posterior portion of the parietal, whereas the lateral dataset PC3 describes variation in the length of the orbits and the height of the skull. PC4 and higher axes each account for less than 5% of variance.

Major clades (i.e. mammals, lepidosaurs, and crocodylians) largely occupy separate regions of morphospace in the dorsal and lateral GM analyses (Figs. 3 and 4), with the dorsal dataset best discriminating clades. Only isolated taxa exemplifying morphological extremes (e.g. *Dasypus novemcinctus*, nine-banded armadillo; highest PC 1 score for mammals) plot as outliers of their clade. Mammal and lepidosaur morphospace adjoin in the short-snouted, large orbit region of morphospace, with the carnivorous phocids *Mirounga*, *Erignathus*, and *Halichoerus* exemplifying this extreme in mammals. In the lateral dataset, crocodylians plot away from the other two major clades, whereas the morphospace of lepidosaurs and mammals overlap to a greater degree than the dataset of skulls in dorsal view, although their ranges remain significantly different (Fig. 4; Procrustes ANOVA $p < 0.001$). Aquatic mammals again display extremes in mammalian skull shape, with select phocids and herbivorous sirenians plotting between a diverse selection of squamates (Fig. 4). Morphological overlap is greatest in the linear morphometric dataset, although taxonomy remains a significant influence (ANOVA $p < 0.001$; electronic supplementary material). Similar to GM datasets, long-snouted crocodylians plot by themselves, but short-snouted taxa (*Paleosuchus* and *Osteolaemus*) overlap with mammals. This overlap also extends to mammals and lepidosaurs and is more pronounced than in either lateral or dorsal GM datasets.

Mammals display a significantly greater amount of morphological disparity (measured as sum of variances) than lepidosaurs or crocodylians in GM analyses (Welch Two Sample t-test $p < 0.001$), in part driven by the remarkably diverse skulls of marine and flying taxa (Figs. 3 and 4, electronic supplementary material). Lepidosaurs have a lower degree of morphological disparity than mammals. In contrast, despite their low species richness compared to mammals and squamates, crocodylians occupy a relatively large range of morphospace, although it is still the lowest of tested clades (electronic supplementary material). It is worth noting that these results are an underestimate of total disparity, especially for mammals and lepidosaurs, as we did not include many groups that would certainly exhibit significant shape diversity, such as cetaceans, primates, and many groups of limbless lizards. In spite of this, we think that these results reflect broad disparity patterns in living amniotes, as we sought to sample taxa that express much of the morphological variation exhibited by their clades.

When phylogeny is accounted for, the relationship between diet and skull shape in the tested GM datasets reveals conflicting assessments (Table 1). In dorsal view, this relationship is not statistically significant ($p = 0.088$), whereas in lateral view it is significant ($p = 0.006$). In both cases, the goodness of fit is low ($R^2 = 0.029$ and 0.044 , respectively), which demonstrates that the predictive power of this relationship is weak. The tests of phylogenetic signal indicate that both dorsal and lateral shapes have less signal than

1 expected under a Brownian motion model of evolution ($K_{\text{mult}} = 0.331$ and 0.394 for dorsal and lateral datasets,
2 respectively) and that variance tends to be distributed more within clades than between clades. When
3 alternative phylogenetic hypotheses are tested the results are not significantly impacted, suggesting the
4 broader phylogenetic relationships are driving this pattern.

[revised manuscript text omitted]

of dental elements [23,30]. These data, as well as the linear analyses here, contain less morphological information, which may help facilitate the overlap found in those studies. Furthermore, the morphological differences driving the separations we observed are not present in unsampled taxa: no marine mammal has enlarged pterygoids comparable to those of the crocodylians in the dataset, regardless of snout length. For these reasons, we predict that these unoccupied regions reflect reality rather than unsampled taxa. Understanding the constraints preventing greater overlap may be a fruitful pursuit for future studies because they may offer insights into the morphological limits of each clade.

Diet has a significant influence on skull shape in the sampled non-avian amniotes in the lateral dataset, but not in the dorsal dataset. Additionally, we do not find an overarching significant predictive relationship between these variables that universally applies across all three clades. Instead, members of the three major clades evolve a variety of skull shapes associated with a given diet that reflects their idiosyncratic combinations of skull architecture, dental morphology, and degree of emphasis on oral processing of food, corroborating findings for a large sample of avians [14]. This result underscores the importance of phylogenetic scale in form-function investigations. Investigations of individual specific clades often demonstrate more distinct relationships with dietary ecology playing a key role in shape [10,12,67,68]. In contrast, when a wider phylogenetic focus is applied, such as in this study, phylogeny tends to overprint trophic patterns, although ecology may continue to play a significant, but less predictive role [5,8,13,14,28,29]. In spite of this result, some degree of convergence is possible, but taking a broad phylogenetic perspective elucidates how general functional principles and specific dietary and other demands map onto particular skull architectures to produce similar or different skull shapes [69].

[revised manuscript text omitted]

Tables

Inserted at the end of the document.

Figures

Figure 1

Figure 2

Figure 3

Figure 4

Figure and table captions

Table 1. Summary statistics of Procrustes ANOVA and phylogenetic ANOVAs testing the relationship between diet, phylogenetic relationships (Procrustes ANOVA only), and skull shape using geometric morphometric shape data on dorsal and lateral datasets. Regressions were run using ‘geomorph’ package.

Figure 1. Representative non-avian amniote skulls used in this study scaled to the same skull length to emphasize proportional differences. A) *Ailurus fulgens* (FMNH 36755) B) *Galerella sanguinea* (FMNH 83649), C) *Sphenodon punctatus* (UF 11978), D) *Dendrolagus lumholtzi* (FMNH 60897); E) *Varanus salvator* (UF 56879); F) *Paleosuchus palpebrosus* (FMNH 69869); H) *Crocodylus niloticus* (TMM M-1786); I) *Gavialis gangeticus* (NHM R. Unnumbered). Abbreviations: FMNH, Field Museum of Natural History, Chicago; NHM, Natural History Museum, London; TMM, Texas Vertebrate Paleontology Collections at The University of Texas, Austin; UF, University of Florida, University of Florida, Gainesville. Scale bar equals 3 cm.

Figure 2. Time calibrated phylogenetic tree of studied non-avian amniotes. Major clades are noted whereas diet is illustrated by terminal branch color. Gray circles represent 100 million years. Silhouettes courtesy phylopic.org and Sarah Werning.

Figure 3. Plot of principle components 1 and 2 for the dorsal geometric morphometric dataset. Clades are represented by shapes and diets with colors. The three major clades plot in separate regions of morphospace with relatively little overlap. Convex hulls outline extent of each clade’s morphospace. Extremes of skull shape along each axis shown using landmarks (anterior to left).

Figure 4. Plot of principle components 1 and 2 for the lateral geometric morphometric dataset. Clades are represented by shapes and diets with colors. Similar to the dorsal dataset, crocodylians plot in separate regions of morphospace, although lepidosaurs and mammals overlap to a greater degree. Convex hulls

outline extent of each clade's morphospace. Extremes of skull shape along each axis shown using landmarks (anterior to left).

Table 1

		Dorsal Dataset							
		DF	SS	MS	R ²	F	Z	P	K _{mult}
Procrustes ANOVA	Diet	3	0.320	0.107	0.069	3.737	3.197	<0.001	
	Residuals	152	4.349	0.029	0.931				
	Total	155	4.669						
	Taxonomy	2	2.131	1.065	0.456	64.183	8.124	<0.001	
	Residuals	153	2.539	0.017	0.544				
	Total	155	4.690						
Phylogenetic ANOVA	Diet	3	0.002	0.0006	0.029	1.564	1.379	0.0882	0.331
	Residuals	152	0.058	0.0004	0.970				
	Total	155	0.060						
		Lateral Dataset							
Procrustes ANOVA	Diet	3	0.300	0.100	0.042	2.129	1.806	0.037	
	Residuals	145	6.813	0.047	0.958				
	Total	148	7.113						
	Taxonomy	2	2.788	1.394	0.392	47.038	7.014	<0.001	
	Residuals	146	4.326	0.030	0.608				
	Total	148	7.114						
Phylogenetic ANOVA	Diet	3	0.003	0.001	0.044	2.207	2.484	0.006	0.394
	Residuals	145	0.075	0.001	0.956				
	Total	148	0.078						

Appendix C

Response to reviewers

Dear RSOS editors,

We would like to thank you for allowing us to revise and resubmit our manuscript “The limits of convergence: the roles of phylogeny and dietary ecology in shaping non-avian amniote crania”. We appreciate the time and the valuable comments provided by the reviewers. We have taken these into account and our resubmitted manuscript includes the suggestions of all reviewers. We believe that their recommendations improve both the clarity and quality of the manuscript. The reviewers’ comments about the methods, specimen selection, and request for additional analyses have all been incorporated into the new draft. Please find below more in-depth responses to individual comments and requests. We hope the reviewers are pleased with the changes. Thank you again for your valuable contributions.

Best,

Keegan Melstrom, Kenneth Angielczyk, Kathleen Ritterbush, and Randall Irmis

Reviewer 1:

“My main concerns have to do with sampling and methodology. Regarding sample, I think the authors can do a better effort to justify the choosing of taxa. I find the exclusion of beaked amniotes poorly elaborated. Clearly, having “heavily modified skulls” could apply also for many toothed amniotes, such as vipers, cetaceans, tapirs, dinosaurs, rodents, dugongs, varanids, etc. In addition, the sample is markedly counterfactual when considering amniote evolution, given is composed of some toothed extant amniotes and excludes not only any fossil taxa but also two major amniote lineages such as birds and turtles. The use of the term “non-avian amniote” is indeed inadequate. In any case, I wonder if an extra emphasis on developing landmarks other than based on teeth could have circumvented this issue and allowed to include some birds and turtles (for example, using the landmark 5 in lateral view to depict also the caudalmost border of the beak, but this is only a suggestion).”

We have revised our explanation of the sampling choices to reflect these concerns. This work represents a substantial portion of the first-author’s dissertation, of which the primary goal was to explore the possibility of using skull shape in a morphometric analysis to help predict diet, adding an additional non-destructive proxy to dietary reconstruction in fossil animals. As such, the purpose was to compare only toothed animals because other proxies (e.g. wear, carbon and oxygen isotopes) are present in those taxa. The past explanation, however, avoided this, which resulted in more questions than answers. We have modified this section to better explain the specimen selection.

The use of the term non-avian amniote was simply meant to be a relatively succinct way of summarizing our dataset. We felt that also mentioning turtles, although very important, would only make the paper more jargon-filled because they have a small fraction of the species diversity of birds. Similarly, extant dentigerous amniotes (here specifying non-beaked animals to the exclusion of fossil taxa) also seemed a little too-jargon heavy. Overall, our primary motivation with non-avian amniote was to summarize our dataset as simply as possible.

“Concerning calculations, I consider that lacking of allometry tests is a major flaw of the analysis. This can be viewed even as a missed opportunity of further examination of the data, given that the authors have had all the elements needed to perform allometric regressions, which can be done straightforwardly from the data using the same software and almost without extra computing effort. Justification of not testing influence of size on the shape and ecology is poor and entirely inadequate, in my view. In addition, without testing allometry, all results displayed should be considered “preliminary”, so testing for allometry could be considered mandatory in this context. It can be the case, for example, that snout elongation has an allometric component, and this could be being completely overlooked. Insectivory is also constrained by body size. I strongly encourage authors to perform allometric regressions on the sample data, to include them as supplementary information, and to discuss it along the manuscript.”

We performed additional ANCOVA and phylogenetic ANOVA tests of the effect of allometry on skull shape as Reviewer 1 requested. We ran these analyses on both the lateral and dorsal GM datasets. Because we had linear measurements for our sampled taxa, we used four size proxies: centroid size (automatically generated by gpagen analyses), skull length, snout length, and skull width, all of which were logged. We find that these tests of allometry are significant in most cases, with some interesting exceptions. We added new tables of results, which is included in our supplementary information, and integrate the results throughout the manuscript. The inclusion of allometry in our analyses did not significantly alter our results and we think only strengthens our arguments. We thank them for their recommendation.

*“Minor improvements could imply explaining more about the calculations, how they were made, which options were chosen (if Euclidean distances, or Bending Energy, was used for sliding semilandmarks, how models –fits– were constructed for ANOVA or what type of SS was computed, for example). The issue is especially important for the ANOVA and how the models were set, shape~diet, shape*diet, etc. R commands accept many different configurations for each analysis, which have impact on results obtained, therefore clarifying this can be very useful for understanding adequately the methodology. Table 1 appears to be mixing Blomberg’s K and phylogenetic ANOVA’s results, which render it confusing.”*

We included some of these new details in the primary manuscript and more specific details on the parameters of R functions are now included in the supplemental information. Additionally, how we constructed the linear models are (e.g., * vs. +) is also explained. We appreciate this request for clarification.

Similarly, we now explain in the caption of Table 1 that the K statistic is a separate analysis from the phylogenetic ANOVA.

(Page 2, ln 48) *“This is a working hypothesis, an assumption, that needs to be tested, or at least addressed again in Conclusions given that could clearly bias the results of the analysis.”*

Yes, it is an assumption, but we would argue a relatively safe one. The absence of teeth in jaw elements significantly alters the developmental trajectory of skull elements as well as the constraints placed on consuming food. Teeth are much denser and stronger than keratin and

bone, and in turn this will permit different strategies for consuming food, which will very likely have an impact on skull shape. Although the absence of beaked taxa has an impact on our results, the same could be argued for any group we leave out. Ultimately, the goal of our work was to look at toothed amniotes to help develop a novel proxy for diet that could be used in conjunction with other dental features to reconstruct diet, in particular fossil crocodylomorphs.

(Page 3, ln 20) *"I find this argument quite weak. Every clade can be defined as "heavily modified" depending on the ancestor to which it is being compared."*

We deleted this clause and have revised our justification for excluding birds and turtles. We chose to focus on toothed taxa early in the research in hopes that skull shape would elucidate aspects of dietary ecology in this group, as had been shown with linear morphometrics in marine taxa. The hypothesis was that skull shape could then be used as an additional, (relatively) independent data point to be compared to aspects of tooth shape, wear, and complexity to elucidate diet for extinct taxa. The ultimate goal of this work was to use skull shape as an additional proxy to reconstruct the diet of extinct crocodylomorphs, many of which have been compared to mammals or squamates. Our choice of these toothed taxa and not other beaked groups was to test if fossil crocodylomorphs overlap with extant mammals and squamates.

(Page 3, ln 43) *"Explain this"*

We have added an additional explanation of the tools used and why.

(Page 3, ln 45) *"The software is actually R, and should be cited properly. Rstudio is only a GUI for R. Indicate also the R packages and tools used to run these calculations and provide their inline citations."*

We have modified the methods to reflect these recommendations.

(Page 3, ln 53) *"Authors can summarize this by saying "to avoid parallax errors".*

Agreed, that is much more succinct.

(Page 3, ln 54) *"This is wrong. Normalization only ensures that size differences between specimens are minimized in order to enhance patterns in other variables. To test influence of size, use allometric regressions."*

We removed the word normalized to better reflect the goal of our analyses.

(Page 3, ln 56) *"In my opinion, this could be considered a major flaw of the analysis. Assessing of allometry (specially in this case, where evolutionary allometry can be expected) is an important step of any morphometric study. Even if the goal is to focus on shape, in many cases shape variation is correlated with size variation, and this should be calculated and discussed. Otherwise, the results depicted are sort of "preliminary" until allometry can be tested, as appears to happen in this study, where potential allometric patterns are dismissed plainly. I*

strongly encourage authors to perform allometric regressions of the Procrustes coordinates again log Centroid Size and including them as supplementary information.”

We appreciate Reviewer 1’s recommendation on this topic. Although it was never our intention to dismiss the effects of allometry, it nevertheless was the outcome of what we had analyzed and written. To address their concern, we have run additional analyses on the geometric morphometric dataset, which make up the bulk of our analyses and results. We measured the effects of allometry in two ways, first by analyzing regressions of the log of centroid size against Procrustes coordinates (as suggested) as well as the log of measured skull length against Procrustes coordinates. In both cases, skull shape is significantly correlated with size. In fact, when size is included as a factor in the phylogenetic ANOVA, diet becomes a significant influence on skull shape, in contrast to when the analysis is run without considering size.

Although we believe these new analyses do not radically change our results, throughout the manuscript we amended our results and discussion to reflect this new information. We thank Reviewer 1 for their recommendation.

(Page 5, ln 54) *“I agree but this is not but a hope”*

True. We have deleted this sentence because it’s out of place in the results and mentioned in the discussion, thereby making it redundant.

(Page 6, ln 2) *“I find this not entirely clear. Is variance within clades larger or smaller than between clades?”*

We have sought to clarify this explanation in the text itself, but to summarize, our dataset implies that there is more morphological divergence within clades that would be expected under a Brownian motion model of evolution. In other words, under BM, the degree of morphological divergence between species is proportional to how distantly related they are, whereas in our dataset there is more variation among closely related species than would be predicted only by branch lengths and topology only. This implies that there is something else causing ‘excess variance’ within clades. That excess variance is likely related, at least potentially, to dietary ecology.

(Page 6, lns 32–35) *“I doubt if this can be included here this given the data patterning was not different from evolutionary noise...” “How it would be limiting convergence if there was not detectable phylogenetical signal?”*

Taking these together because they’re addressing the same basic concern in our manuscript. We would disagree with this assessment. We show that evolutionary history plays a key role in our dataset. Most broadly, in our Procrustes ANCOVA analyses we find that phylogenetic history is a significant influence on skull shape in both the lateral and dorsal datasets. In fact, in the lateral dataset, phylogeny is demonstrated to be a better predictor of cranial morphology than diet (Table 1). Secondly, and more importantly, when phylogenetic relationships are explicitly taken into account the significance of diet decreases, clearly demonstrating the importance and impact that phylogenetic history is having in our initial results.

(Page 6, ln 44) “*I find this a truism. Clearly if any given animal is assigned to a lineage, it is because the basic pattern of the clade can be recognized despite actual modifications...*”

We believe that in the context of our study, assessing broad patterns of variation across a large group of living animals, that this statement is not necessarily devoid of importance. Additionally, our future work exploring extinct crocodylomorphs seeks to address the idea that many fossil taxa were ‘mammal-like’. In some cases, previous authors were referring to dental occlusion, but others imply that the skull shapes have converged on mammal morphologies.

(Page 7, ln 20) “*But phylogenetic signal was non-significant!!! K is closer to 0 than to 1...*”

I think this is the source of a lot of our misunderstanding. Yes, K is closer to 0 in our GM analyses (0.33 and 0.39 in dorsal and lateral, respectively), but K is measuring expected phylogenetic signal under Brownian motion. So our results indicate that there is less phylogenetic signal in our dataset than expected under a scenario in which morphological divergence is exactly proportional to branch lengths, and our values of K are actually significant ($p = 0.001$) when compared to randomized data (i.e. the true case of no phylogenetic signal). Furthermore, although our phylogenetic signal is low, our phylogenetic ANOVA results indicate that it does have a significant effect on skull shape. Given the distance between our clades (~325 million years), we are not surprised that there is a significant deviation from what is expected under a Brownian-motion model. In tests of phylogenetic signal, Revell et al. (2008) found that low phylogenetic signals, including values near what we observe here, can be produced by a wide variety of evolutionary scenarios. In fact, in those scenarios, punctuated divergent selection tends to lead to lower phylogenetic signals. Not that we’re saying that’s happening in our dataset, Revell et al. (2008) were careful to point out that low phylogenetic signals can be produced by wide ranging effects. For this reason, we avoid assigning drivers for the low value.

(Page 7, ln 25) “*Or they retain short snouts?*”

Given the fossil record, which includes many long-snouted relatives, we are confident in saying that neither of these groups retain a short-snouted condition form.

(Page 7, lns 26–29) “*If comparable muscle architecture is present*” “*For instance, cranial kinesis?*” “*again, cranial kinesis and secondary palate, elaborate further on these topics*”

Thank you for the recommendations. Where commented on, these have recommendations have been added.

(Page 7, ln 34) “*In spite of having long snouts, which appear to contradict the previous claiming, unless the authors elaborate a bit more on the subject.*”

We have sought to clarify our meaning here. We appreciate you pointing out this because it did seem like a contradiction. Overall, the point of this section is to point out that although diet, in many cases (but not all) has a significant influence on skull shape, there are often many ways to achieve an “ideal” shape for each diet. We have convergence in some cases, such as the short snouts of some carnivorous lizards (e.g. *Heloderma*) and mammals (e.g. *Felis*), but we also have

noteworthy differences, in this case varanids. They are a successful group of (mostly) predators that don't utilize high bite forces like many other carnivores. They're a great example of the fact that there are many "solutions" to ecological "problems", which is why we emphasized them as an example. Although interestingly they do tend to converge on smaller crocodylians, which appear to be a bit more terrestrial than other, larger crocodylians and take terrestrial prey, similar to varanids. This, however, was a bit too speculative to include in the MS because many of these smaller crocodylians tend to be a bit more cryptic and so the dietary data is not as thorough as other taxa, like the American Alligator.

Reviewer 2:

"My only real question was around whether some of the gaps in morphospace between major clades might disappear if fossil taxa were included, and so are perhaps the result of extinction. I am particularly thinking of the much more diverse skull morphologies present in fossil crocodylomorphs, which include short-snouted forms. While I do not think the authors necessarily need to modify their paper, a brief consideration of this point in the discussion might be worthwhile."

As we mentioned in our response to Reviewer 1, this contribution represents a large proportion of a dissertation that also investigates fossil taxa. We are happy that Reviewer 2 was interested in this, but we felt that any significant mention of fossils may take away from the primary focus of the manuscript, extant forms. Stay tuned!

Reviewer 3:

1) I'm more familiar with stratigraphic ways of time-calibrating supertrees, so if the below is not 100% relevant I apologise. You mention throughout the importance of identifying phylogenetic signals in your dataset to interpret phylogenetic constraints in skull morphology and convergence. How come you then use only one time-scaling method? Phylogenetic signals can be influenced by branch lengths and inferred divergence dates and therefore it may be prudent to use an additional independent (or two if possible) time-scaling method and test for phylogenetic signal. Examples employed by many palaeobiologists include FDB, hedman, mbl and cal3. With the exception of FDB these time-scaling methods can be easily done in R. If there are similar levels of phylogenetic signal present in a second time-scaled tree then this would show your results are not a false positive from your chosen time-scaling method."

We only used one type of time calibration method because many of the nodes are relatively well resolved. When we built our tree in Mesquite, we scaled it by hand. Additionally, unless we are mistaken, many of the suggested time scaling functions utilize stratigraphic ranges, which are not applicable here because all taxa included are extant (with the exception of Stellar's sea cow, which is recently extinct). Many of the time-scaling methods appear to require ranges for taxa include and calculate node times based off of those.

"2) You mention in your results: "When alternative phylogenetic hypotheses are tested the results are not significantly impacted, suggesting the broader phylogenetic relationships are driving this pattern" but don't elaborate further. Where are the results for these alternative

phylogenies? I understand that you used some different mammal relationships from Upham et al. but did you use completely different supertrees when testing for alternative phylogenies? You're not incorrect in the way that have resolved polytomies by adding 1 million years to such polytomies and to unresolved divergence dates, but say you had a three way polytomy with taxa A, B and C, how did you decide which taxon of the three diverges first? The best way to account for polytomies and unresolved nodes would be to create 10-20 randomly resolved versions in R (using one of the time-scaling methods mentioned in point 1) and perform the phylosignal analysis (just the K_{mult} or lambda statistic, see point 3) on all of them. If the trees have the same strength of phylogenetic signal then you can proceed with one tree (i.e. the one of you already have) for all other analyses."

We did not report on the results of these alternative hypotheses in the original manuscript. In total, we examined three different phylogenetic arrangements. We report on the tree that is general consensus of the sources of each supertree (Upham for mammals, Pyron for lepidosaurs, and Erickson for crocodylians, the latter of which isn't a supertree). In the second tree, we rearrange following alternative phylogenetic positions of mammal taxa reported by Upham et al. (2019), including the arrangement of macropodids and phocids, but the position of major clades (e.g. carnivorans, rodents) were not changed. Much of the variation in these trees is due to differences in taxa or subclades not included in our dataset. The final tree also modifies the arrangement of squamates in a more significant way, rearranging the position of tropidurids, crotaphytids, and liolaemids. When physignal is run on these data, the difference between K_{mult} value is in the thousandths. This information has been added to the supplemental information

"3) The lambda statistic is also good for deducing levels of phylogenetic signal in a dataset (Munkemuller et al. 2012; Meth. Ecol. Evol. 3, 743-756). This can be performed using the phylosig package and function of the same name in R. This would provide an independent test of whether PC 1 and PC 2 values for the dorsal and lateral data exhibit phylogenetic signals for the entire dataset."

We have taken your suggestion and run the phylosig analyses. Interestingly, we found a low ($\lambda = 0.388$) value for our PC1 results, which is similar to our Blomberg's K statistic, in which we also found a lower phylogenetic signal than expected under Brownian Motion, although the λ result was just outside of statistical significance ($p = 0.078$). Additionally, we found effectively no phylogenetic signal for our PC2 data. We now include this information in our supplemental information, thank you for the suggestion!

(Page 1, ln 11) *"Terminology gripe. You mention your focus is on skulls but when we get to the methods and results you actually focus on the cranium. The skull is the cranium + mandible and so where relevant please replace skull with cranium as this may confuse some readers."*

Thank you very much, we have changed skull to crania (or some variation of it) throughout the manuscript.

(Pages 1–2) *"Would this be better phrased as "...exhibit extensive diversity within their respective regions."?", "Perhaps include a reference that cites the split between synapsids and diapsids occurred 300+ Mya", "exhibit rather than express"*

Recommendations accepted.

“What do you mean by well-documented? I.E. feeding observations, stomach contents?”

Well-documented means that diets are typically described by either direct observation or more than one source. In taxa that are well known, there is often a fair bit of dietary variation based on season, food availability, or individuals. So by drawing from a greater number of sources we were able to get a better picture of diet, when possible.

(Page 3, ln 9) *“So I noticed that the smooth-helmeted iguana is something of an outlier among lepidosaurs in your PC datasets. I googled it and they have this really cool casque. Is this structure due to sexual selection? Would such structures have confounding effects on your dataset?”*

Yes, *Corytophanes* is the major outlier for all of squamates due to its crest. The prominent crest on *Corytophanes* is not due to sexual selection and is present in both males and females, although males tend to have slightly larger crests. When choosing our dataset we tried to avoid taxa with major differences in morphology between sexes to avoid issues associated with sexual dimorphism. In most cases, however, we do not think there would be significant differences. We did investigate individual variation in reduced datasets (Fig. S4) and there was rarely major differences between individuals.

In response to you last question, we will go even broader than sexual selection structures, would large crests have confounding effects on our dataset? Yes, but because these crests are bony and not cartilaginous they represent the full spectrum of possible variation and should, therefore, be included. In the case of *Corytophanes*, the crest is an outgrowth of the parietal and it represents an extreme case, but illustrates what is possible in squamate morphospace.

(Page 3, ln 11) *“Did you make sure to only sample adults? I assume you did but best to say as such.”*

Yes, we have added a sentence in the methods clarifying this.

(Page 3, ln 21) *“I didn’t check through your species list exhaustively, but did you include fossorial lizards? Do they have the same problems as snakes?”*

No, we did not include fossorial lizards for the reason you point out – they are missing key landmarks. Although different ones than snakes, in the case of fossorial taxa the limits of the orbit are often difficult to repeatedly place.

(Page 3, ln 34) *“It may be worthwhile having a simple version of your Supp. Figs. 1-3 in the main text so that readers immediately know what data you’ve measured. E.g. Use the *Crocodylus* from each figure as a part a, b and c. Happy to be over-ruled on this however.”*

We have added a figure illustrating the GM and LM landmarking schemes using *Crocodylus* as suggested. Thank you for the recommendation.

(Page 3, ln 60) *“I’m assuming that your initial supertree had more taxa than what you included in your final dataset? If so, state the number of taxa you originally had in the supertree and then how these taxa were dropped to result in only your sampled taxa.”*

The supertrees that our trees were based off of did have (many) more taxa. The mammal tree had 5,911 species (Upham et al., 2019), the squamate tree had 4,161 species (Pyron et al., 2013), and the crocodylian tree had 22 species (Erickson et al., 2012). When we generated the tree, we took only the topology for the taxa included in our dataset from the underlying supertrees to build ours in Mesquite.

(Page 4, ln 10) *“Be careful how you phrase this because anteater is the common name of a species in Mammalia.”*

Thank you, we have reworded this section to be more precise.

(Page 4, ln 11) *“Assigned would be a better word then placed.”*

Recommendation accepted.

(Page 6, ln 54) *“Why do you think the Dasypus plotted where it did? Is it due to fossorial adaptations? Consumption of formicids? Is it the only xenarthran in your dataset which do have “weird” skulls?”*

The placement of *Dasypus* is very likely due to a combination of traits that, by themselves, would not have placed them within crocodylian morphospace but together become ‘croc-like’. The most prominent trait is the elongate snout, but also its narrowness combined with the distally placed orbits are the features that separate crocodylians and *Dasypus* from other long-snouted mammals.

(Page 6, ln 56) *“I think you're right in that diet and phylogeny are the main determinants of skull size. However, have you thought about the role of biotope and what taxa are doing in these biotopes. For example, crocodylians are characterised by their dorso-ventrally flattened snouts (even the brevirostrine taxa) and this come in handy for ambushing prey above the water. Do phocids show extremes because they are simply in a different biotope to much of your other mammal taxa (and thus have different selection pressures related to locomotion)?”*

Thank you for this suggestion! Because we were not testing for the effects of specific environments in our dataset we did not mentioned them in detail. Regardless, these are important influences on skull shape that should definitely be mentioned, especially because many outliers, like you pointed out, are marine taxa. We have added in a few sentences discussing the importance of these functional considerations in both mammals and crocodylians in the discussion, although slightly above where you noted.

(Page 7, Ins 32,33) “Cite *McCurry et al. (2015) PLoS ONE 10, e0130625 here*” “I don;t think the PGLS reference is suitable here. I would use *D'Amore & Blumenschine (2009) Paleobiology, 35, 525-552*”

There was an error in the reference software for the PGLS paper! Thank you for catching this, it has been corrected and additional references have been added.

(Fig. 2 [now 3]) “1) *Remove the black circles apart from the nodes you've labelled (Lepidosauria, Iguania etc.). This would make some of the more cluttered clades look less cluttered.*

2) *The dashed lines around some clades looks untidy. Could you remove these and replace them with curved lines around the edges of the fan. E.g. Fig. 1, Field et al. (2018) Curr. Biol. 28, 1825-1831.*”

Both of these recommendations have been followed.

“I know that the dots are the semi-landmarks but they are a little unclear in representing skull shape diversity. I know that this one is a crocodilian but that's because I'm familiar with then. For the others I'm not too sure. A possible solution may be to have a line schematic of the skull with the semi-landmarks imposed over it so that readers can more easily connect the shapes to the points on the figure? This would need to be done for all main text and supplementary figures that includes these landmarks.”

The landmark schemes don't necessarily represent actual skulls, only the maxima and minima of each landmark position on the PC axis. What that means is that there isn't a precise skull we can point to or draw that represents each image. We have updated our figure caption to more accurately reflect what is being depicted

Reviewer 4:

“My primary concern is that I found somewhat confusing the way the Result section was organised. In particular, the distinction between the Results of the Dorsal-view, Lateral-view GM and linear morphometrics analyses is not particularly clear. I often found myself wondering whether the authors were referring to one or another. Were the dorsal and lateral datasets analysed separately in the GM? And was this distinction maintained also in the Linear measurement? If not, how were the linear measurements compared to either the dorsal or lateral datasets? Some of these details are in the Supplementary Information but should really be clearer in the main text.”

We have sought to clarify these issues throughout the main text. In particular, in the methods we make explicit that we analyzed three separate datasets and explain what those are. The dorsal and lateral datasets were analyzed separately, as was the linear morphometric dataset. Throughout the results and discussion, we sought to provide more detail about what dataset in particular we are referring to. Additionally, in the results we moved the linear morphometrics results to the end of the section. We chose not to break up the results of the two GM datasets. Many of the results are

similar, such as the primary shape drivers of the first PC axis, and we felt that the section read more smoothly with these datasets being directly compared rather than broken up into distinct sections. The discussion, however, has been split up into distinct sections now.

“How did comparisons between GM and Linear morphometrics worked (if it was quantitatively done)? It should be explicitly specified which datasets were compared. Perhaps, more importantly, is the distinction between the GM and linear morphometrics sections. These should be clearer to the reader. A potential solution could be adding sub-sections in the Method and Result section and discuss them separately.”

Comparisons between the GM datasets were a mix of quantitative and qualitative. Comparisons of ANCOVA and phylogenetic ANOVA results were quantitative, whereas those of location of specific taxa or groups on PC axes or interpretation of morphology of PC axes was qualitative. Similarly, comparisons between GM and LM datasets were qualitative only, although we avoid direct contrasts except for an initial broad comparison.

It is very confusing that the Result section that there is barely mentioned in the linear morphometric dataset. What happened to it?

We have added significantly to the linear morphometrics results section. Additionally, we have also moved the reporting of this information to the bottom of the results section, better delineating it from the results of the GM analyses.

“The Discussion section would also, in my opinion, benefit from being split into 2/3 sub-sections with appropriate headings. It would simply be a matter of moving around some of the paragraphs and expanding some others accordingly. A lot of interesting topics are mentioned, and I think it would strengthen the paper if they were discussed in more detail separately. For instance, I would love to see more group-specific (both extinct and extant) and biomechanics studies be discussed with the results of this manuscript analyses. This is somewhat done briefly in places, but I think that focusing a bit more on the biomechanics would be beneficial for the discussion (e.g. the role of size).”

We have followed this recommendation and added three sub-headings to the discussion as well as repositioned some paragraphs as needed. Additionally, we have also included biomechanical studies at particular points, included mention of cranial kinesis and the bite force of *Varanus*. Aside from these, we chose not to add significant comments on biomechanical implications of our results, primarily because this paper is intended to set up a framework for adding in fossil taxa in future studies and it felt as though discussion of biomechanics was outside the scope of our original intent.

“SUPPLEMENTARY INFORMATION

The formatting of references to Figures and Tables and Supplementary Information is not consistent within the main text and between Main text and Supporting Information.

Large parts of the Supplementary Information text (e.g. Statistical analyses) are the exact copy of what is written in the main manuscript. I am not sure why, but some parts do not add anything

to what said in the main manuscript. Please carefully revise both documents to avoid repetitions and use the supplementary file for supporting information!"

We have fixed the formatting errors and modified the portions that were exact copies. We apologize for both of those oversights. Additional information that was present in the supplemental information, such as the semilandmarking procedure, has also been added into the main text at the request of other reviewers. We also added important information on the process we used to modify our trees and new phylogenetic ANOVA results. Lastly, the summary statistics of all new allometry analyses have been incorporated into supplementary tables.

“FIGURES

I do like the figures as they are -they are attractive, informative and immediate. I was wondering if the landmarks and measurements figures could be merged with figure 1 of the main text. Landmark and curve numbers should be added to them.”

See below.

“OTHER COMMENTS

Have you thought to run a Linear discriminant analyses and test how well the PC coordinate of all datasets (Linear morphometrics and GM) predict dietary assignments? In my experience with a variety of Jurassic marine reptiles (ichthyosaurs, plesiosaurs, thalattosuchians) it won't work really well for predicting inferred diet but would assign well taxa to phylogenetic groups (which is perhaps hardly surprising as we use a lot of cranial characters do define different lineages). This is perhaps something that could be added to the Discussion?.”

We had considered running Linear Discriminant Analyses earlier in the process, but ultimately decided against it. Our results generally agree with your predication and previous experience, with phylogenetic groups being the more prominent influence on skull shape, more so than diet. We ultimately chose not to run LDA because our current analyses showed that using skull shape by itself to predict diet was ultimately not reliable and the LDA would likely have been redundant with that result.

“Make sure that you adequately reference all the software that had been used.”

All software is now referenced.

(Page 1, ln 57) *“Examples, references? And is this a problem? If so, please explain the reason why.”*

This line has been deleted because it was a bit repetitive from what was stated below. We added in important references to the lower paragraph. To answer your second question, no, this is definitely not a problem and is not intended to be a criticism of previous researchers, simply a justification for our broader study. Earlier, targeted studies that focus on smaller clades are extremely important and allow for high-resolution patterns to be observed and discovered.

(Page 1, ln 59) *“or rather feeding strategy/behavior?”*

If we were comparing major clades to each other, we would certainly include these things, but because in this hypothetical example (canids vs. crocodylians) the strategies are so different (active hunting on land vs. ambush predator at the land-water interface) as well as oral processing behavior (mastication vs. swallowing whole) we originally thought that it was the diet and its functional requirements that would be driving shape.

(Page 2, ln 1) *“Do you have any reference for this?”*

Added!

(Page 2, lns 10–12) *“This repeats the previous paragraph a little. Please revise.”*

We deleted the previous line to avoid repetition.

(Page 2, ln 30) *“again is this diet or feeding behaviour? I am not sure myself, but a sentence of clarification about this should be added.”*

Previous researchers have referred directly to diet or dietary niche. However, we see your point how calling it diet itself may be inaccurate and overly simplistic. We added a sentence clarifying that the mechanical properties of food are likely the driver of many cranial changes.

Additionally, we modified diet to dietary ecology, which is a broader term throughout the manuscript.

(Page 2, ln 38) *“please find references for these concepts too.”*

References added.

(Page 3, ln 11) *“If possible please add brackets here with the specific groups that were excluded. I know they are listed later, but having all of them here would be helpful.”*

The major clades were added.

(Page 3, ln 28) *“Fig. 2 has nothing to do with this statement. Please revise.”*

In the previous draft it was mislabeled, but with the addition of a new figure this label is now correct.

(Page 3, ln 30) *“please specify first how many datasets you used then explain what they are.”*

A sentence has been added that explains the requested information.

(Page 3, ln 33) [In reference to the electronic supplementary information] *“format.”*

This section has now been formatted.

(Page 3, ln 34) *“I would prefer these to be illustrated in the main manuscript, but I understand if there are space constraints. The main landmarks should be appropriately numbered (they are not in the current figures in the esm). This will help with replicability.”*

Both Reviewer 3 and 4 requested a figure illustrating the GM and LM landmarking/measurement schemes. We added a new figure (Fig. 2) that illustrates these key features. Additionally, we modified the electronic supplementary figures of the GM landmark schemes to label each landmark, with an explanation of the abbreviations provided in Table S2.

(Page 3, lns 43, 45) *“please reference this software”*

FIJI was referenced (Schindelin et al., 2012), but R was not. Both are now cited.

(Page 4, ln 7) *“how many and which? This information should be explicitly written at least in the supporting information.”*

See comments to Reviewer 3 for more in depth response. In short, we generated two alternative trees, with modifications in the arrangement of phocids, marsupials, and iguanids in Mesquite. There was no significant difference in phylogenetic ANOVA results or the strength of phylogenetic signal, suggesting our results are mostly driven by broad phylogenetic relationships that are widely agreed on. We have added the information about the results of these two analyses in the supplemental information.

(Page 4, ln 37) *“Separately? Which datasets were compared exactly? Lateral GM ; Dorsal GM and Linear morphometrics? Please be more specific, this information is essential to ensure repeatability.”*

All datasets were analyzed separately, so PCAs, Procrustes ANCOVA, and phylogenetic ANOVA were performed on the two sets of geometric morphometric data independently. Similarly, the linear morphometric data were analyzed separately. We have rewritten the beginning of this paragraph to make this more clear.

(Page 4, ln 37) *“you should be more specific here. Evidently you refer only to GM datasets in this paragraph: please explicitly say so, otherwise it will be confusing for readers that are not familiar with these techniques.”*

Correct, we are referring to the GM tests first, then the LM. We added a short explanation to the second sentence that clarifies this.

(Page 4, ln 39) *“reference to “geomorph” package is missing.”*

We added the reference.

(Page 4, ln 54) *“or you could have used the PC scores and run a LDA testing how well the PC scores predict a priori assigned dietary categories (and doing the same for phylogenetic groups). Have you tested that linear morphometrics data and GM data gave the same results?”*

The linear morphometric workflow was modeled off of Jon Mitchell's research investigating the disparity of extant and extinct birds, which also utilized linear morphometrics and allowed the combination of discrete and continuous data (Mitchell and Makovicky, 2014; Mitchell, 2015). An LDA would have been another great option. Both LM and GM data return significant differences between dietary and taxonomic datasets. The LM data, however, showed a lot of overlap between diets and, with a few exceptions, is not very useful for dietary reconstruction.

(Page 4, ln 55) "*punctuation and/or capital letter issue*"

Thank you for catching that!

(Page 4, lns 55–62) "*I think I understood this but this is a little hard to follow. Please try simplify this.*"

We added more explanation of the motivation for the Gower distance step and tried to clean up the writing.

(Page 5, ln 9) "*I wonder if it is worth subdividing the results section into: GM and Linear Morphometrics.*

The latter is barely mentioned here - and makes it all a bit confusing."

We have taken your advice and broken the GM and linear morphometrics into two sections, with the linear morphometrics section having been expanded and placed at the end of the results. We chose to not emphasize the linear morphometrics results for two primary reasons: 1) they contain less shape information than the GM results and 2) phylogenetic relationships seem to be driving the major patterns we observed and we were not able to remove the effect of phylogeny from the CCA results. Additionally (and not as critical), the linear morphometrics data were not easily visualized, aside from the figure shown in the supplemental information. For these reasons we chose to emphasize the results of our GM analyses, but we still thought that it was important that we do report our linear morphometrics results.

(Page 5, ln 55) "*this explanation should also go in the method section*"

Deleted and mentioned in methods.

(Page 6, ln 27) "*what about patterns within crocodylians? there is plenty of literature describing them - do your result agree (broadly) with what was previously found?*"

All crocodylians are considered carnivores in our dataset, which is why we do not detail their patterns in this section. Our results are very similar to those of previous researchers that have looked at crocodylian skull in GM analyses. In particular, our results closely follow those of both Pierce et al. (2008) and Wilberg (2017).

(Page 6, ln 40) “*But there is no explanation of the linear dataset in the results. What happened to it?*”

As noted above, we chose to emphasize the results of our geometric morphometric dataset because those data contained more detailed shape information, could be visualized more clearly, and we were better able to tease apart differing effects of diet, allometry, and phylogeny from that dataset. Basically, we felt that drawing conclusions from the GM data were safer than the LM, but thought it was important to report both.

(Page 6, lns 50–51) “*Could this be expanded a little? especially with reference to the fossil record: many alleged cases of morphological convergence occur in marine tetrapods (ichthyosaurs, pliosaurids, thalattosuchians, with cetaceans).*”

We have added a bit about this. However, we have an upcoming manuscript that investigates patterns in extinct crocodylomorphs that will expand on this further.

(Page 7, lns 30–31) “*True: size also plays a role in here. There are a bunch of studies on extant and extinct taxa that shows that very large animals (e.g. pliosaurids) do not have specific adaptations to optimise bite force, because their large size itself does the trick. Something on this line may be worth mentioning. So this is a complex situation with many factors in play.*”

We expanded this section to include references to recent studies on varanid bite forces, but we did not add a mention of size. From what we have been able to find, it seems as though all measured varanids have relatively weak bite forces, even *Varanus komodoensis*, which is an exceptionally large taxon.

(Page 8, ln 1) “*you mean a "less extreme" morphology?*”

Yes, we have modified our phrasing and followed your recommendation.

(Page 8, ln 12) “*dietary categories?*”

Yes, but we were trying to avoid being redundant with the ‘dietary ecology’ that was mentioned earlier in the clause.